# Single cell RNA sequencing identifies early diversity of sensory neurons forming via bi-potential intermediates

Louis Faure [1], Yiqiao Wang[2], Maria Eleni Kastriti [1], Paula Fontanet [2], Kylie K. Y. Cheung[2], Charles Petitpré[2], Haohao Wu[2], Lynn Linyu Sun[2], Karen Runge[3], Laura Croci [4], Mark A. Landy [5], Helen C. Lai [5], Gian Giacomo Consalez[4], Antoine de Chevigny [3], François Lallemend[2,6], Igor Adameyko [1,7] & Saida Hadjab [2✉]

Somatic sensation is defined by the existence of a diversity of primary sensory neurons with unique biological features and response profiles to external and internal stimuli. However, there is no coherent picture about how this diversity of cell states is transcriptionally generated. Here, we use deep single cell analysis to resolve fate splits and molecular biasing processes during sensory neurogenesis in mice. Our results identify a complex series of successive and specific transcriptional changes in post-mitotic neurons that delineate hierarchical regulatory states leading to the generation of the main sensory neuron classes. In addition, our analysis identifies previously undetected early gene modules expressed long before fate determination although being clearly associated with defined sensory subtypes. Overall, the early diversity of sensory neurons is generated through successive bi-potential intermediates in which synchronization of relevant gene modules and concurrent repression of competing fate programs precede cell fate stabilization and final commitment.

[1] Department of Molecular Neurosciences, Center for Brain Research, Medical University Vienna, 1090 Vienna, Austria. [2] Department of Neuroscience, Karolinska Institutet, Stockholm, Sweden. [3] INMED INSERM U1249, Aix-Marseille University, Marseille, France. [4] Università Vita-Salute San Raffaele, 20132 Milan, Italy. [5] Department of Neuroscience, UT Southwestern Medical Center, 5323 Harry Hines Boulevard, Dallas, TX 75390, USA. [6] Ming-Wai Lau Centre for Reparative Medicine, Stockholm node, Karolinska Institutet, Stockholm, Sweden. [7] Department of Physiology and Pharmacology, Karolinska Institutet, 17177 Stockholm, Sweden. ✉email: saida.hadjab@ki.se

In the developing nervous system, various neuronal and glial cell types are generated from multipotent progenitor cells in a specific chronological sequence. The continuous change in the cellular potential of the progenitors is a key aspect to this developmental process[1]. As neuronal progenitors exit cell cycle, they differentiate into distinct neuron types that already provide the underpinning for the identity of neuronal circuit in which they will operate[1,2]. Hence, the formation of specific fates critically depends on the timing and diversity of expression of molecular factors in discrete parts of the nervous system[3,4]. However, following this process, how the expression of molecular determinants is progressively acquired and restricted to particular neurons as they further diversify into distinct neuronal classes remains unclear. A holistic understanding of the stepwise execution of elaborate transcriptional programs for neuronal diversification in heterogeneous system is still elusive. It is therefore essential to dissect and understand the precise temporal progression of the gene-regulatory networks that produce and assemble neuronal complexity. Indeed, such complexity is found in all parts of the developing nervous system, including the highly heterogeneous population of sensory neurons—cells that live in dorsal root ganglia (DRG) and provide us with sensations of touch, pain, itch, temperature and position in space[5].

The diversity of sensory modalities emerges during early embryonic development as subtypes of sensory neurons are generated from a common progenitor population, the neural crest cells (NCCs). Two main waves of sensory neurogenesis in DRG occur between E9.5 and E13. The first wave of neurogenesis (E9.5−10.5)[6] gives rise to myelinated neurons, the cutaneous low-threshold mechanoreceptors (here after named mechanoreceptors) and the muscle proprioceptive afferents (here after named proprioceptors) and a small population of Aδ-fibers that can be distinguished by E11.5 based on their specific expression of transmembrane receptors and transcription factors[6,7]. Mechanoreceptors express RET and MAFA and/or TRKB, proprioceptors express tropomyosin receptor kinase C (TRKC) and RUNX3, and the small Aδ-nociceptor population expresses TRKA. Later, these populations will diversify even further, giving rise to additional subtypes representing the medium-to-large diameter DRG neurons responsible for muscle proprioceptive feedback and skin mechanosensation modalities[5]. A second wave of neurogenesis (that peaks at E12 in a mouse embryo) generates the small diameter unmyelinated nociceptive neurons[5–7], which co-express TRKA and RUNX1 and later will diversify into subtypes necessary for pain, itch and thermal sensation[8] and innervate the skin and deep tissues. Although the heterogeneity of DRG neurons in adult mice are increasingly characterized[8,9], the logic and molecular mechanisms that allow progenitors and sensory neurons to fate split and unfold specific sensory neuronal subprograms from sensory-biased NCCs remain to be understood.

Here we use deep single-cell RNA sequencing (scRNAseq) together with other experimental approaches to investigate fate choices, lineage relationships and molecular determinants during early stages of sensory neurogenesis in the mouse. Overall, our data provide insights into the structure of the Waddingtonian landscape of the sensory lineage and suggest that counteracting gene-regulatory networks operate within immature postmitotic neurons, resulting in the emergence of alternative-lineage transcriptional programs defining the neuron fate choice.

## Results

**Differentiation trajectories and major fate splits**. To obtain a map of lineage splits from progenitors towards specific sensory subtypes during mouse embryonic development, we sequenced the transcriptome of individual single cells (progenitors and neurons) of the somatosensory neuro-glial progeny of the trunk. For this, we FACS isolated tdTomato positive (TOM$^+$) cells isolated from mouse lines which selectively represent all NCCs ($Wnt1^{Cre}$;$R26^{tdTOM}$ and $Plp1^{CreERT2}$;$R26^{tdTOM}$) and from neuron-specific Cre mouse lines ($Isl1^{Cre}$;$R26^{tdTOM}$ and $Ntrk3^{Cre}$; $R26^{tdTOM}$) to obtain an enrichment of somatosensory neuronal populations (Supplementary Fig. 1a, b) from E9.5, E10.5, E11.5 and E12.5 and sequenced mRNAs of single cells with high coverage using Smart-seq2 protocol (median of 8070 genes detected per cell) (Fig. 1a and Supplementary Fig. 1c, d; see "Methods"). The selected stages correspond to key events of sensory lineage development starting from migrating NCCs (E9.5) and finishing with the end of the second wave of sensory neurogenesis and the early specification of neuronal subtypes (E12.5). Our dataset contains cells from $Isl1^{Cre}$;$R26^{tdTOM}$ at E9.5 and E10.5 and $Ntrk3^{Cre}$;$R26^{tdTOM}$ at E11.5 and E12.5, both lines label sensory neurons, $Wnt1^{Cre}$;$R26^{tdTOM}$ was used to label NCCs, glial and neuronal progenitors at E9.5 and E10.5 and $Plp1^{CreERT2}$; $R26^{tdTOM}$ at E12.5. The combination of cells collected after tracing with these Cre lines should provide a complete compendium of neuronal progenitors and early neuronal subtypes in mouse DRG (for FACS gating strategies, see Supplementary Fig. 2). Co-embedding of the cells from all tracing strategies was made without tracing-based batch correction and lead to overlapping of cells of different tracings following both a developmental time-wise trajectory and a progression from progenitors to specified neurons (Fig. 1a–e and Supplementary Fig. 1a, b; see below for trajectories) which together convincingly validated the use of the dataset for further computational analysis.

We used RNA velocity[10], an unbiased approach that leverages the distinction of spliced and unspliced RNA transcripts from the aligned sequences, allowing us to obtain an additional timepoint for each cells ($t$: expression of old spliced RNA and $t + 1$: expression of new unspliced RNA). Using the proportion between these two values for a given gene, and under some assumptions, it is possible to infer whether the expression of a gene is being initiated or downregulated. By combining this information for all genes in each cell, as well as comparing it to its neighbor cells, it is possible to infer a direction vector indicating the putative future transcriptional state of the cell. In our dataset, RNA velocity revealed a clear differentiation directionality from NCCs to neuronal progenies. In order to recapitulate such transitions, the dataset was first projected onto diffusion space to denoise the underlying geometry, and a principal tree has been fitted using ElPiGraph[11] in a semi-supervised way with the help of clustering results (Fig. 1b, c). ElPiGraph is a manifold learning method which aims at inferring a principal graph (such as a tree) "passing through the middle of data" in high dimension. Cells were ordered along the principal tree, and for each cell the distance on the graph to a chosen root is considered as a pseudotime value. Pseudotime analysis of gene expression showed significant and robust changes along the reconstructed tree (8432 genes at a false discovery rate (FDR) of 0.05), leading to the identification of major clusters (Source Data file). It also captured tree root populations and endpoints as well as intermediate states as subclusters, leading to the identification of two main neurogenic trajectories named branch A and branch B (Fig. 1c). Those branches evolved from a clear transition from progenitors to post-mitotic newborn neurons. This was determined using Pagoda2 (Fig. 1d, e)[12], which separated the lineage tree into two major states, cycling ($Sox10^+$ and $Sox2^+$) versus non-cycling ($Isl1^+$), corresponding to non-neuronal versus neuronal populations, the latter being defined by specific expression of axon-related cytoskeleton and function genes (including $Tubb3$, coding for βIII-tubulin) (Fig. 1d, e). Consistently, SOX10$^+$ cells

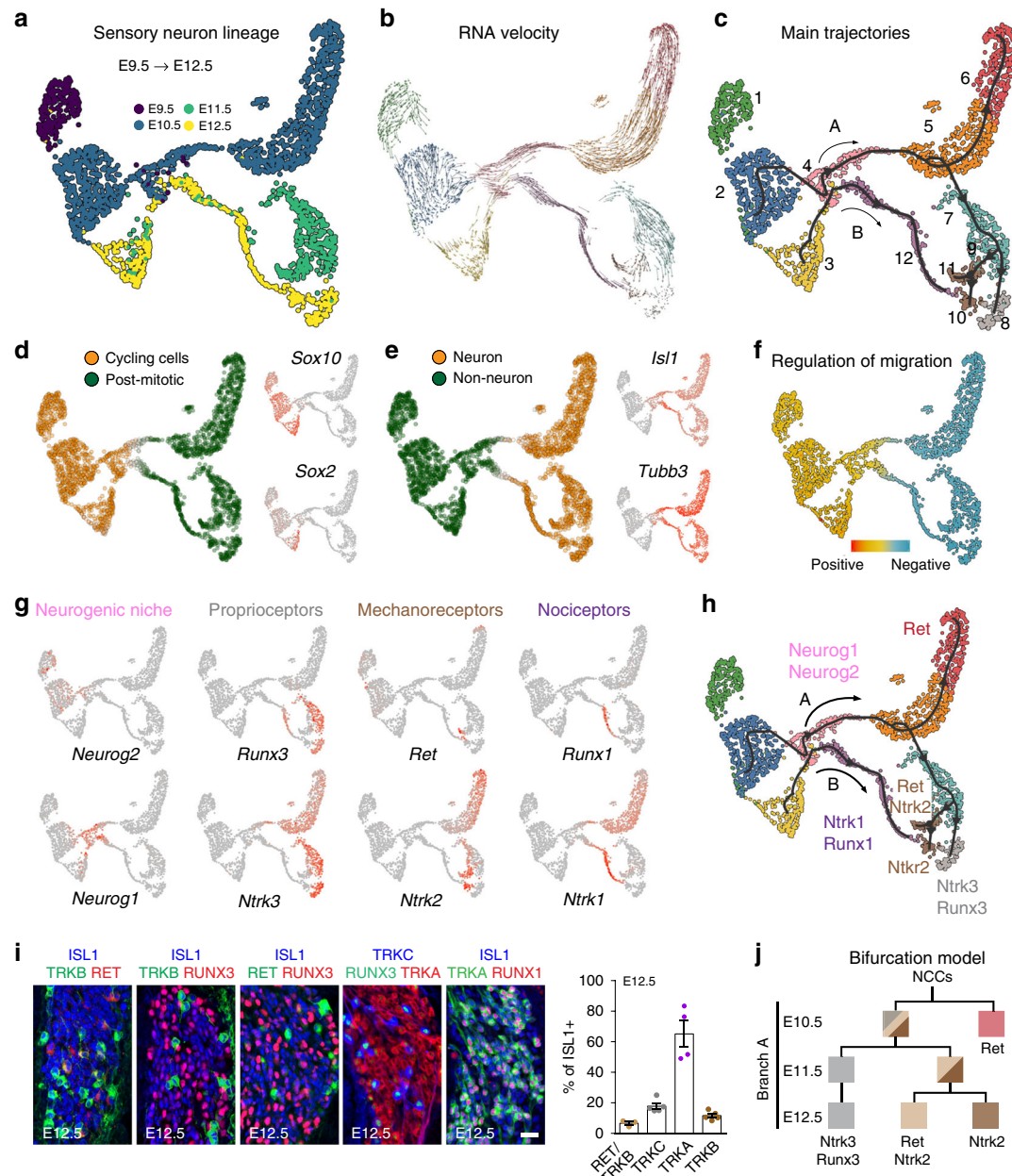

**Fig. 1 scRNAseq and pseudotime analysis of the developing somatosensory system. a** UMAP embedding representation of the single-cell RNA sequencing dataset, annotated by embryonic day. **b** RNA Velocity vectors projected onto the UMAP embedding, indicating differentiation directionality. **c** Differentiation trajectories inferred in a semi-supervised way on Diffusion space using ElPiGraph, revealing 12 main clusters represented within two main trajectories (branches A and B), 10 branches and 3 bifurcations (1, 2 and 3) on branch A. List of genes is provided in the Source data file. **d**, **e** Most significant biological aspects extracted using pagoda2 indicate cell state changes from cycling ($Sox10^+$ and $Sox2^+$) to post-mitotic cells (**d**) and from non-neuronal to neuronal cells (**e**) ($Isl1^+$ and $Tubb3^+$). **f** PC score from gene set related to GO term "positive regulation of cell migration" (GO:0030335) subtracted by the PC score for "negative regulation of cell migration" (GO:0030336) indicates a transition from migratory neural crest progenitors to settling neuronal populations. **g** UMAP plots of selected genes distributed along the trajectories and among cells states represented in (**c–e**). **h** Transcriptomic dynamic during neurogenesis and neuronal specification show that the different states account for the differentiation of sensory subclasses that can be distinguished based on their specific expression of neurotrophic factors receptor and transcription factors ($Ntrk1$, $Ntrk2$, $Ntrk3$, $Ret$, $Runx1$ and $Runx3$). **i** E12.5 DRG sections immunostained for markers highlighted in (**h**) representing major sensory subpopulation at the trajectories endpoints and quantification ($n = 3-5$). Scale bar, 20 μm. Data are presented as mean values ± SEM. **j** Hierarchical bifurcation model of NCCs-derived sensory neurons differentiation within Branch A based on our scRNAseq data analysis (color code and cluster identity according to panel **h**). Mixed color squares reflect the potential fate choice that the lineage retains at the corresponding developmental point.

incorporated EdU (cycling cells) and did not co-localize with the post-mitotic neuronal marker ISL1 (Supplementary Fig. 3a). Similarly, βIII-tubullin staining labeled sensory neurons and did not co-localize with SOX2+ cells, which were also EdU+ (Supplementary Fig. 3b). PC scores for positive (GO:0030335) and negative (GO:0030336) regulation of cell migration GO term gene sets showed a transition from migratory NCCs to neuronal progenitors (Fig. 1f). Interestingly, as NCCs migrate and coalesce into sensory ganglia in vivo, they showed enriched expression of genes associated with interactions with the local

microenvironment, including integrin subunits, the principal receptors for binding and interacting dynamically with the extracellular matrix, which represents a necessary property of motile cells (Supplementary Fig. 3c). In contrast, neuronal progenitors and neurons were defined by a switch in expression of cadherin genes that promote cell-to-cell adhesion (Supplementary Fig. 3d), which is important for cell positioning and border delimitation at a tissue level. Thus, these findings reveal a general principle and molecular pathways governing the change in cell adhesiveness associated with migrating precursors and post migratory neuron progenitors.

Within the cycling root populations, some cells expressed the pro-neural basic helix loop helix transcription factors *Neurog1* and *Neurog2* (coding for neurogenins 1 and 2, NGN1 and 2), identifying the neurogenic niche prior to their differentiation into post-mitotic sensory neurons. From these progenitors, branches A and B differentiated to endpoint neuronal clusters that express by E12.5 the molecular markers characteristic of the main early DRG neuronal populations: the proprioceptive, mechanoreceptive and nociceptive populations. The proprioceptive lineage is characterized at E12.5 by the co-expression of *Ntrk3* (*Ntrk3* coding for TRKC) with *Runx3*[13–15] while the mechanoreceptive neurons express *Ret* and *Ntrk2* (*Ntrk2* coding for TRKB)[5,16] (Fig. 1g–i and Source Data file). The mechanoreceptors further branches into *Ntrk2*+ only cells and *Ret*+/*Ntrk2*+ cells, which constitute subpopulations of this lineage with distinct connectivity and function in the adult[5,17,18]. Altogether, those populations represent branch A. In contrast, branch B differentiated into nociceptive *Ntrk1*+ (TRKA) cells and is characterized by the expression of *Ntrk1* (and the beginning of induction of *Runx1* in the TRKA+ cells) (Fig. 1g–i), a specific marker of the nociceptive lineage[5]. Combinatorial use of the above cited population markers covers all (ISL1+)[19] sensory neurons at this stage (Fig. 1i).

Hence, by representing the main early sensory neuron categories, our dataset and analysis generated a refined order of the somatosensory neurons diversification tree whereby progenitors undergo rapid transcriptional changes that progress through stepwise branching points and discrete sensory neuron states consistent with a developmental ordering of the emergence of the different neuron types (Fig. 1j). This refined map helps in understanding the temporal sequence and lineage relationship along the differentiation trajectories of the main somatic sensory neurons observed at this stage and will allow further explorations on the molecular principles underlying their generation.

**Stem cell populations and waves of sensory neurogenesis.** During neural crest migration, sequential pools of progenitors depend on either NGN2 or NGN1 to generate large-size neurons (mostly TRKC+, TRKB+ and RET+ neurons) of the first wave, then small-size TRKA+ neurons of the second wave of neurogenesis, respectively[20]. A third wave of neurogenesis arises from boundary cap cells (BCCs, which derive from NCCs too) and generates mostly small-size TRKA+ neurons[21]. One major question in the field is how cells progress from one state to another during neurogenesis and how neurogenic programs specify the two major waves of neurogenesis leading to specific neuronal subtypes. To address those questions, we first identified the different clusters of cells and trajectories. In our dataset, the clusters 1–3 (cycling cells), which define the origin of the differentiation tree, showed progressive downregulation of the remaining neural plate border genes (e.g. *Zic* and *Msx*, found in only few cells of cluster 1), upregulation of the NCCs specifiers (*Sox9*, *Foxd3* and *Ets1*) and expression of genes involved in NCCs migration (*Cdh11* and *Itga4*, the latter being highly expressed in

the bridge between clusters 2 and 3) (Fig. 2a, b). Also, following the RNA velocity stream, clusters 2 and 3 started to express pro-neurogenic markers (*Neurog1* and *Neurog2*) (Fig. 2c), defining those populations as neuronal progenitors. While transcriptionally close to cluster 2, the cluster 3 was marked by the expression of genes specific of the BCCs (*Egr2*, *Ntn5*, *Prss56*, *Hey2* and *Wif1*)[22,23]. We therefore labeled the three clusters as early NCCs (cluster 1, top differentially expressed genes (TDEGs): *Prtg*, *Crabp1*, *Snai1* and *Dlx2*), late NCCs (lNCCs, cluster 2, TDEGs: *Hoxd10*, *Hoxd11*, *Hoxa11* and *Hoxc12*) and lNCCs/BCCs (cluster 3, TDEGs: *Fabp7*, *Sparc*, *Serpine2* and *Ednrb*) (Fig. 2a, b, Supplementary Fig. 4a and information on the BCCs in cluster 3 found in Source Data file).

The alignment of the sequencing data revealed two trajectories (or branches) emerging from the highly heterogeneous clusters NCCs and lNCCs/BCCs and named hereafter branch A and branch B leading to the onset of neurogenesis (Fig. 2c). Our analysis also indicates that while neuronal progenitors from lNCCs/BCCs expressed only *Neurog1*, the two main neurogenesis trajectories (branches A and B, Fig. 2c) could not be distinguished based on their specific NGN expression. Indeed, in both branches, progenitors evolved from a *Neurog2*+ to a *Neurog1*+ state (Fig. 2d) and numerous single cells were captured expressing both transcripts simultaneously (Fig. 2e–g). These results contrast with previous data suggesting specific expression of NGN2 and of NGN1 within the first and second waves of neurogenesis, respectively[20]. To confirm our results, we performed genetic cell-lineage tracing analysis of the NGN1 and NGN2 expressing progenitor cell lineages using *Neurog2*^CreERT2^;*R26*^tdTOM^ and *Neurog1*^Cre^;*R26*^tdTOM^ mice. In both mice, recombination (TOM+) was observed in all classes of DRG neurons, regardless of their neurogenesis wave (Fig. 2h–l)[24,25], confirming results from a recent study[26]. Hence, we conclude that most sensory neuron progenitors sequentially express *Neurog2* then *Neurog1* with co-expression in an intermediate state where both genes are co-expressed and suggest that the neuronal precursors generating the first or the second waves of neurogenesis are timely poised to preferentially require either NGN2 or NGN1 for neuronal differentiation.

**Neurogenesis of myelinated and unmyelinated neurons.** We next investigated the genesis and key molecular determinants of myelinated versus unmyelinated neuronal differentiation. To this end, we used diffusion maps to computationally compare the two neurogenesis trajectories, i.e. branches A and B (Fig. 2m). A common point representing the position in diffusion space where the two branches were at their closest was inferred (Fig. 2n). Comparison was performed between the two branches starting from this inferred point. This analysis revealed the modular pattern of gene expression along the trajectories. We could identify specific clusters of co-expressed genes uniquely distributed along or shared between each trajectory and which constituted divergent and convergent genes, respectively (Fig. 2o, p and Source Data file). Comparing our results with the known regulatory interactions between pro-neural genes from previous studies[25], we identified a cascade of TFs expressed in each branch and which are likely to act in the acquisition of a neuronal fate. Hence, after *Sox10* expression, *Neurog2* was followed by *Neurog1*, then *Neurod4*, *Pou4f1* (BRN3A), *Neurod1* and *Isl1*. Convergent genes common to the two main trajectories defined a generic sensory differentiation program (Fig. 2n). The common genes included *Neurod1* and *Neurod4*, which are conserved pro-neurogenic transcription factors and *Sox2* and *Sox5*, which represent previously undescribed findings and might play a specific role in the development of the somatosensory system (Fig. 2p and Supplementary Fig. 2). At the same time, divergent genes

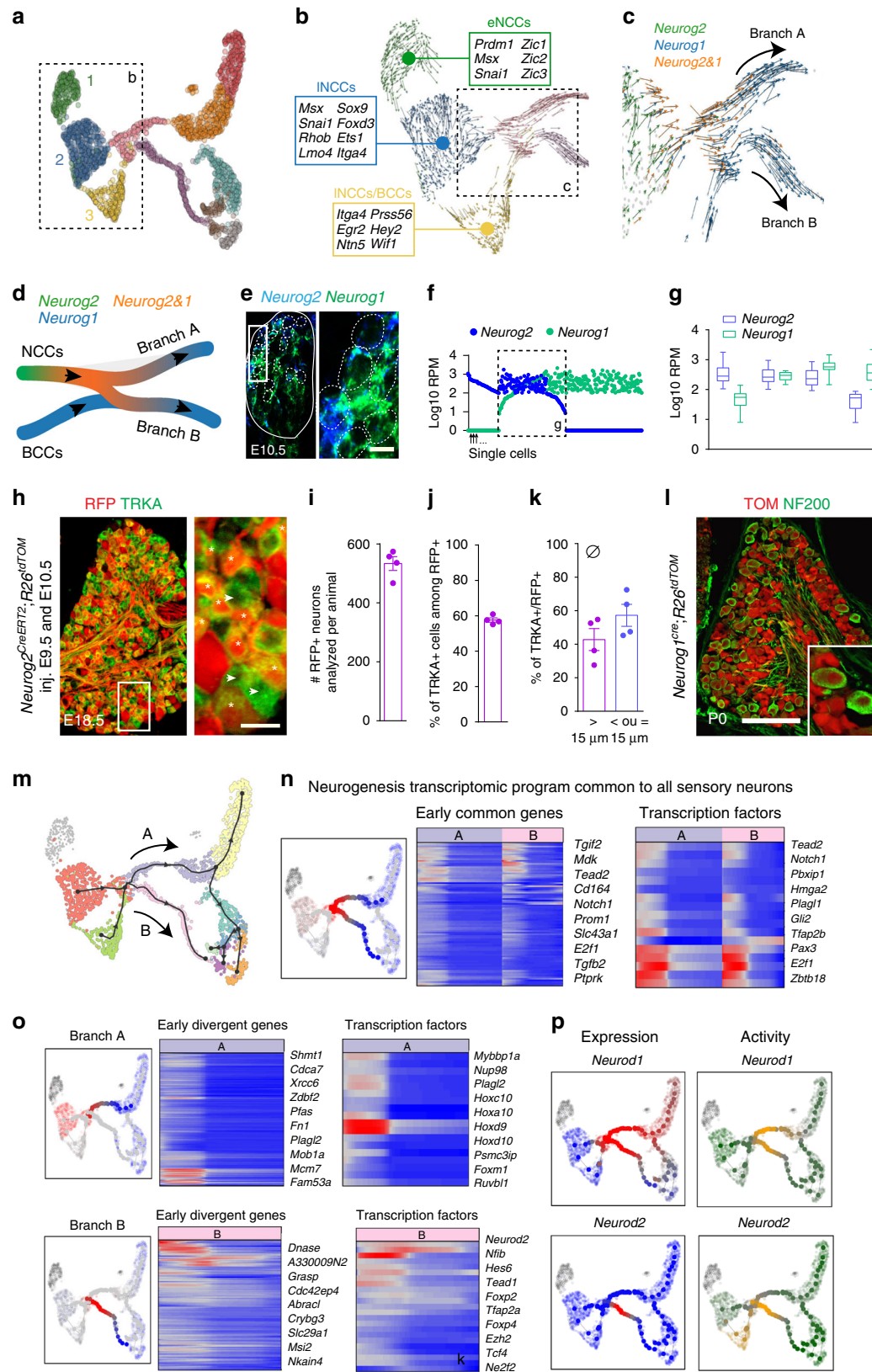

specific to either trajectory distinguished the myelinated (branch A) from the unmyelinated lineage (branch B) (Fig. 2o). Specifically, we found the involvement of *Nfia*, *Gli2* and *Neurod2* in branch B (Fig. 2o and Supplementary Fig. 2), suggesting critical role of distinct TFs during sensory neurogenesis of branches A and B. Overall, differentiation of myelinated versus unmyelinated

neurons proceed via different intermediate states demarcated by the expression of TFs, chromatin modifiers (*Prdm12* as previously shown[25] and *Prdm8*) and signaling molecules. Presumably, the dynamic local signaling landscape might influence and bias the neuronal progenitors towards either branch A or B. Finally, separate GO term analysis on each of the gene sets (common and

**Fig. 2 Neural crest stem cell populations and waves of sensory neurogenesis. a, b** NCCs heterogeneity is defined by different clusters (1−3) (**a**) of cycling cells (Fig. 1d) which are marked by specific expression of markers and delineate early neural crest cells (eNCCs), late NCCs (lNCCs) and boundary cap cells (lNCCs/BCCs) (**b**). RNA Velocity in (**b**) shows the directional transcriptomic flow from eNCCs to lNCCs, then to lNCCs/BCCs. Some cells from lNCCs and lNCC/BCCs converge to the neurogenic state (in pink). **c, d** Analysis of the neurogenic niche with RNA Velocity computation showing sequential expression of the main neurogenic transcription factors *Neurog2* and *Neurog1*. **e** RNAscope staining for *Neurog1* and *Neurog2* on E10.5 DRG sections confirms their co-expression in the same progenitors. Scale bar, 10 µm. **f** Plot of single cells values for *Neurog1* and *Neurog2* shows the existence of three stages among progenitors following the pseudotime at E10.5, including concomitant expression of *Neurog2* and *Neurog1* at the single-cell level (within dashed lines, 136 cells). **g** Quantification of the concomitant average expression among neuronal progenitor cells reflects a dynamic range of expression from high to low *Neurog2* expression and low to high expression of *Neurog1*. Data are presented as Min to Max whisker plots with center point as mean. **h** Cross-section of E18.5 DRG from *Neurog2^CreERT2;R26^tdTOM* injected at E9.5 and E10.5 with tamoxifen shows recombination in neurons originating from the two waves of neurogenesis (branches A and B), as shown by expression of TOM in large diameter neurons and in small diameter TRKA positive neurons (asterisks) (arrows point to TRKA$^+$/RFP$^-$ cells). Scale bar, 20 µm. **i–k** 533 TOM$^+$ cells were analyzed per animal (**i–k**, *n* = 4). Among the RFP$^+$ neurons, more than half were TRKA positive (**j**), with a diameter inferior or equal to 15 µm (**k**). Scale bar, 50 µm. Data are presented as mean values ± SEM. **l** Similar to (**h**), using *Neurog1^Cre;R26^tdTom* (*n* = 3). **m** Identification of common transcriptomic program expressed between branch A and branch B leading to all sensory neurons (**n**). **o** Specific gene modules for the generation of branch A neurons (myelinated, large diameter) or branch B neurons (unmyelinated, small diameter). **p** Transcription factor activity inference shows predictive branch-specific activity.

branch specific) for cellular components showed concomitant enrichment for intracellular part (GO:0044424, GO:0005622). This, in line with the previous conclusion, implies that these waves of neurogenesis are characterized by drastic reconfiguration of the internal components of the progenitor cells.

**From gene modules to specific cell fate choice**. Next, we questioned how the emergence of the main neuronal subtypes is dictated at the transcriptomic level around the two last, most downstream, hierarchical fate split points in our dataset. For this, we examined the course of transcriptional changes and the branching points representing neuronal subtype diversification events, focusing our analysis on the three downstream bifurcations along branch A (Fig. 3a). Branch A gives rise to three mechanosensitive cell types, namely the RUNX3/TRKC proprioceptors and the two mechanoreceptors subpopulations TRKB/RET and TRKB only. Although the identity of the sensory neuron clusters observed at E12.5 is consistent with previous knowledge in the field[5], our aim was to explore the early genes possibly driving fate choice and fate biasing before actual fate commitment.

Each bifurcation could be broken down into three intermediate stages: pre-bifurcation (root), bifurcation point and post-bifurcation part of the trajectory (Fig. 3b). The pre-bifurcation and post-bifurcation stages reveal the dynamic of transcriptional changes leading to fate choice and to its consolidation, respectively. We identified gene modules (groups of genes that change in the same direction and tend to synchronize along the pseudotime) that we qualified as early and late and which characterized the pre- and post-bifurcation stages of the first bifurcation respectively and therefore corresponded to fate choice and fate commitment. Within branch A, the first bifurcation corresponded to the binary cell fate choice between mechanoreceptors and proprioceptors (Fig. 3b, c). We identified two main early gene modules associated with the fate split towards either proprioceptor (45 genes) or mechanoreceptor (60 genes) fates at the pre-bifurcation stage (Fig. 3d) and late genes modules that define cell commitment (Fig. 3e, Supplementary Fig. 4 and Source Data file). Interestingly, before the bifurcation node, the pattern of expression of those gene modules changed dynamically along the differentiation trajectory. Indeed, as cells progressed on the pseudotime axis, the two gene modules were found originally co-expressed in single cells and then gradually mutually exclusive towards the bifurcation point correlating to preference in fate choice (Fig. 3d). GO term analysis for cellular component showed enrichment in postsynaptic membrane (GO:0045211) and synapse (GO:0045202) for mechanoreceptors fate early gene

module. Intrinsic components of membrane (GO:0031224) and membrane parts (GO:0044425) are the top hits for the early gene module of the proprioceptors. Compared to transcriptomic hallmarks of the neurogenesis waves, this suggests that remodeling in post-mitotic immature neurons likely affects the protein composition of the external parts of the cells, presumably for differential interactions with the environment. Other genes are related to cytoskeleton remodeling, including *Cdh4, Pcdh9, Dock5, Dock3* for the proprioceptors (mutation of *Dock3* is known to results in ataxia) and *Stom* and *Pdlim1* for mechanoreceptors, as well as ion channels (*Asic1* and *Kcnip2* for proprioceptive fate).

We next analyzed the inference of cell fate decision governed by TF activity and found TFs differentially expressed for both fates, where *Runx3* and *Nfia* are pro-proprioceptors and *Pou6f2, Nr5a2, Hoxb5, Egr1* and *Junb* are pro-mechanoreceptors (Fig. 3d and Source Data file). Interestingly, *Nfia* and *Nr5a2* were previously described in other systems to be linked to cell fate decision during neuronal development[27] and *Runx3*, a main regulator of proprioceptive neurons development[13,28], was one of the two TFs that belong to the early gene module of the pro-proprioceptor trajectory. Hence, our data support a role for RUNX3 in cell fate choice decision well before the bifurcation point between proprioceptor and mechanoreceptor neurons and suggest co-activation of competing programs prior to fate commitment.

The second bifurcation in the A-fibers low-thresold mechanoreceptors differentiation trajectory represented the fate choice between *Ntrk2* only and *Ntrk2/Ret* (Fig. 3f–h). In adults, those two populations are further diversified and contribute to touch sensation and are distinguished by discrete innervation patterns of cutaneous end organs and different electrophysiological properties[29]. Our data could identify early and late gene modules defining their fate split biasing fate towards either *Ntrk2* only or *Ntrk2/Ret* populations (Fig. 3i, j). GO term enrichment analysis revealed enrichment in axon guidance receptor activity (GO:0008046) for *Ntrk2* only fate. Similar to the previous bifurcation, *Ntrk2/Ret* early gene module was enriched in plasma-membrane-related genes (GO:0005886).

Altogether, along the trajectory following the pseudotime, we identified bursting of modules of highly specific and coherent genes. Although limited at this point to a transcriptomic analysis, these data suggest a competition between these early modules within the early cells of the myelinated lineage that would likely result in biasing the future fate split towards either mechanoreceptors or proprioceptors (bifurcation 1, Fig. 4a) (or towards either two different subpopulations of tactile mechanoreceptors,

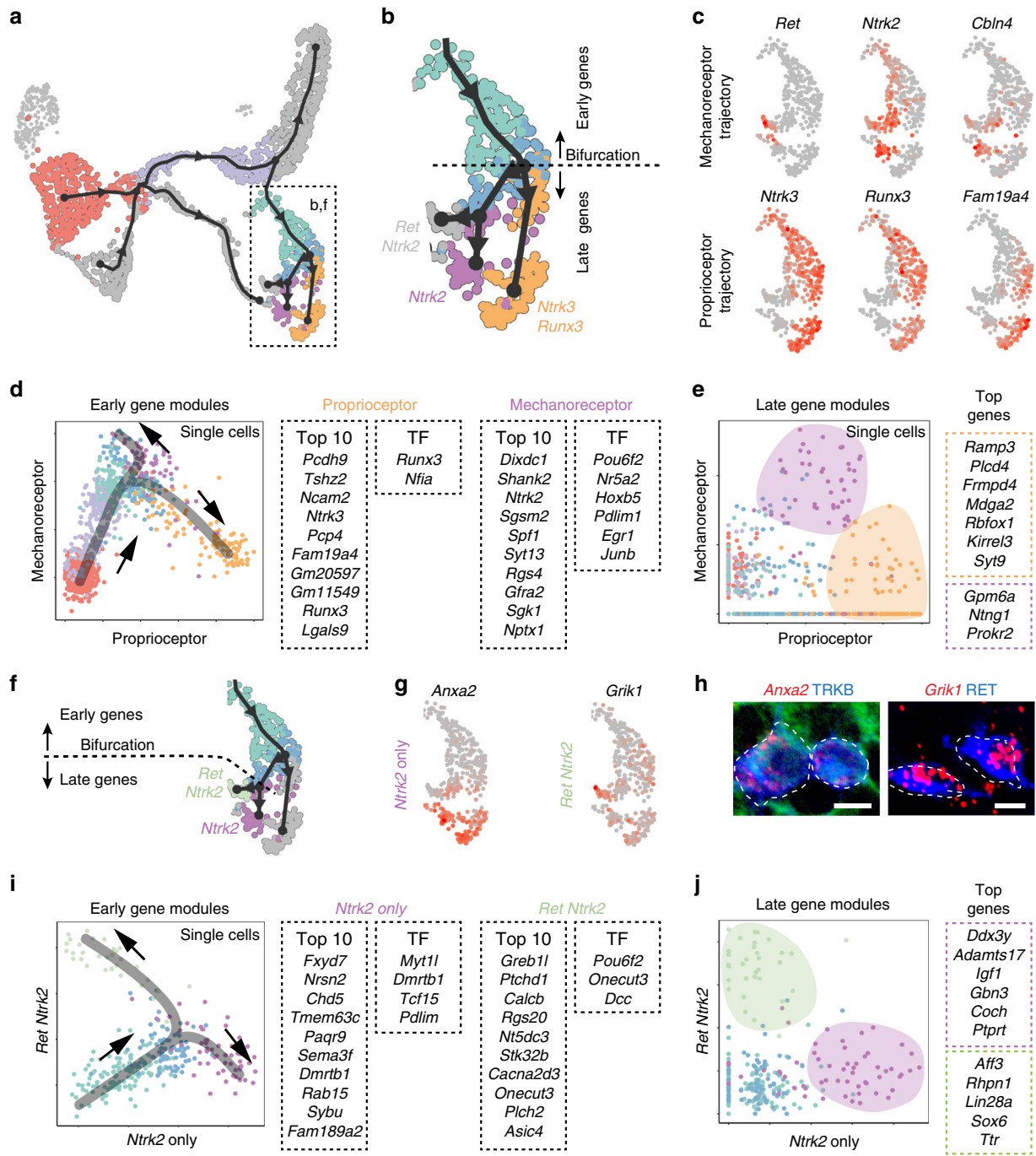

**Fig. 3 Expression of gene modules defining fate choice and commitment along the differentiation trajectories. a** Overview of the analyzed bifurcations on UMAP embedding. **b** Analysis of the bifurcation representing fate choice between proprioceptor and mechanoreceptor lineages (branches 8 and 9 of Fig. 1c). **c** UMAP plots showing expression of markers for mechanoreceptor and proprioceptor lineages and validated in vivo (see Supplementary Fig. 3). **d, e** Scatter plots show average expression of mechanoreceptor and proprioceptor modules in each cell along the mechanoreceptor and proprioceptor lineages. Early competing modules (**d**) show gradual co-activation, followed by selective upregulation of one fate-specific module and downregulation of the alternative fate-specific module. Late modules (**e**) show almost mutually exclusive expression within the proprioceptors and mechanoreceptors after bifurcation reflecting commitment to either fates. Color codes as in (**a**). Top ten highest differentially expressed genes are shown, as well as transcription factors (TF). **f** Analysis of the bifurcation representing mechanoreceptor fate choice between *Ntrk2* and *Ret/Ntrk2* lineages (branches 10 and 11 of Fig. 1c). **g, h** In vivo validation of the branches, with expression of *Anxa2* for *Ntrk2* fate and *Grik1* for *Ret/Ntrk2* fate ($n = 3$ animals). Scale bars, 10 μm. **i, j** Scatter plots show average expression of *Ntrk2* and *Ret/Ntrk2* modules in each cell among the two mechanoreceptor sub-lineages. Early competing modules (**i**) show gradual co-activation, followed by selective upregulation of one fate-specific module and downregulation of the alternative fate-specific module. Late modules (**j**) show almost mutually exclusive expression within the two mechanoreceptors lineage after bifurcation. Color codes as in (**a**). Top ten highest differentially expressed genes are shown, as well as transcription factors (TF).

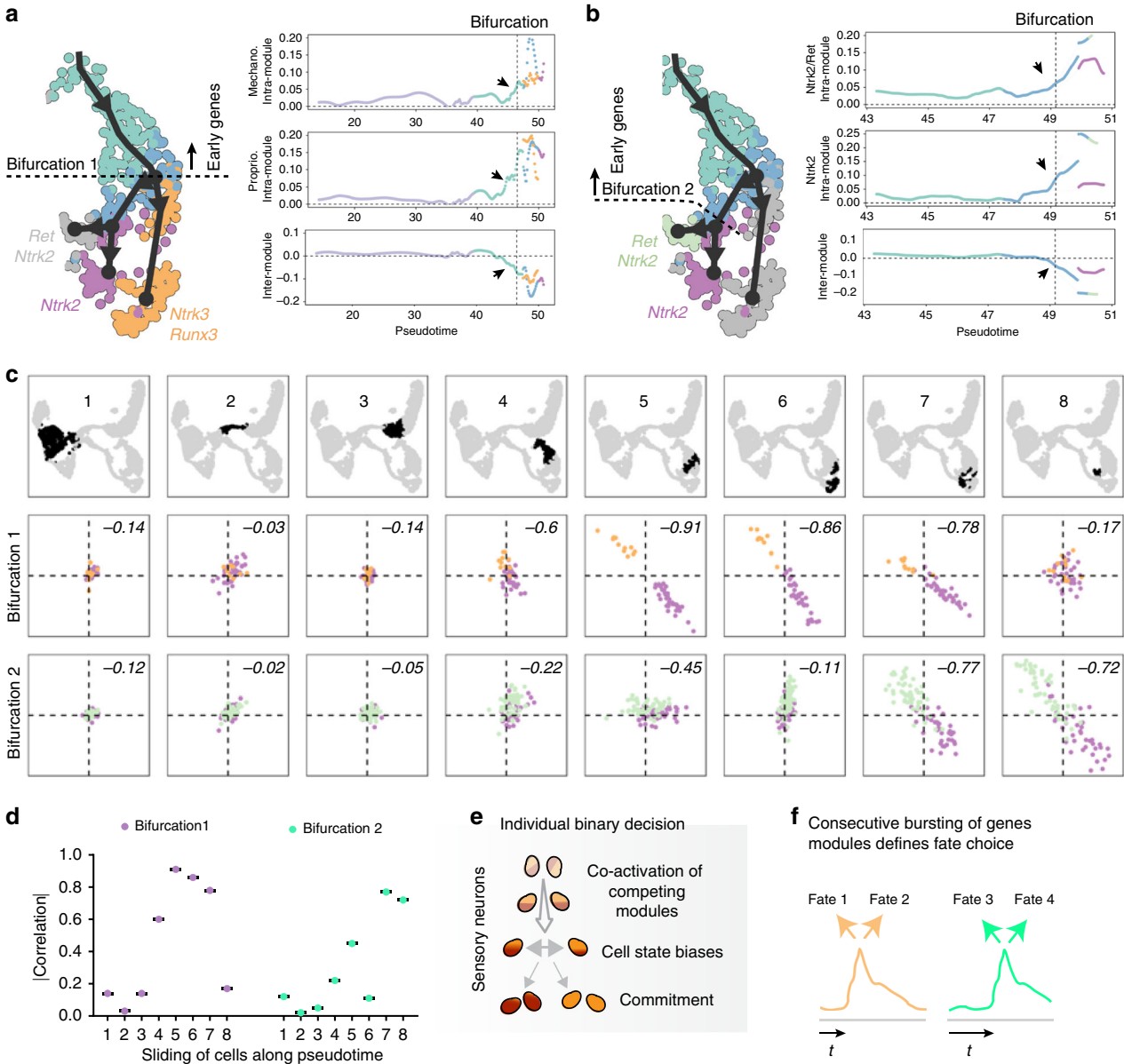

**Fig. 4 Early fate-determining genes or modules show gradual intra-module coordination and inter-module repulsion during cell fate selection. a, b** For bifurcations 1 and 2, plots in a pseudotime analysis show average local correlations of genes within and between branch-specific modules, with gradual increase in coordination within each module and increasing antagonistic expression between the branch-specific modules. **c, d** The plots show average local correlations of module genes with branch-specific early gene modules for both bifurcations, in a set of cells with similar developmental time (marked by black dots). The difference between intra- and inter-module correlations, which characterize the extent of the antagonistic expression between the alternate modules, is shown in the upper right corner of the correlation plots, and represented in a graph in (**d**) as absolute value in Y axis which would reflect the repulsion between modules. **e** Schematic of binary fate decision in sensory neurons. **f** Repulsion of gene modules for the two bifurcations peaks at different pseudotime *t*.

bifurcation 2, Fig. 4b) before the actual split occurs (Fig. 4a, b). Those early biasing modules appear initially positively correlated locally at the single-cell level before the bifurcation (Fig. 4a, b, intra-module, see arrows), and present an increase of the negative correlation inter-modules at the pre-bifurcation stage (Fig. 4a, b, inter-module, see arrows, Fig. 4c), hence suggesting co-activation of competing biasing programs prior to fate commitment (Fig. 4c–f). As a result, the retained program would participate then in the unfolding of the fate towards one cell type versus the other.

**Cell fate-biasing programs in vivo**. To investigate how TFs among early gene modules can affect fate choice in vivo, we

on the role of RUNX3, which was found expressed in part of the sensory neurons within the pre-bifurcation stage of the proprioceptor/mechanoreceptor trajectory and in all proprioceptors after bifurcation (Figs. 3c, d, 5a, b). Using $Runx3^{-/-};Bax^{-/-}$ animals, in which the deletion of *Bax* allows the study of RUNX3 function in the absence of cell death[28], we observed a threefold increase in the proportion of TRKB[+] cells in E12.5 DRG (Fig. 5c), similar to previous results[13]. This suggests a fate change toward mechanoreceptor fate in the absence of RUNX3. To investigate further a potential cell fate change, we mapped the genes composing the early modules that we previously identified within the proprioceptor trajectory onto a published differential gene expression analysis of (FACS isolated and RNA sequenced) *Runx3* null and $Runx3^{+/-}$

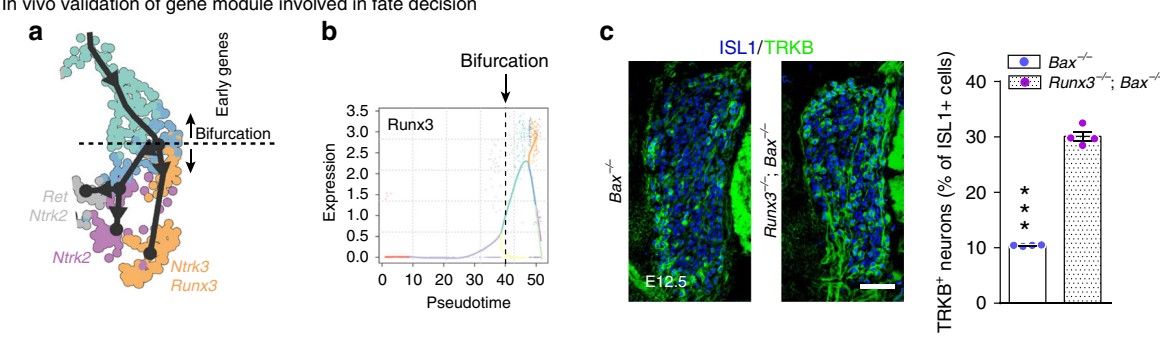

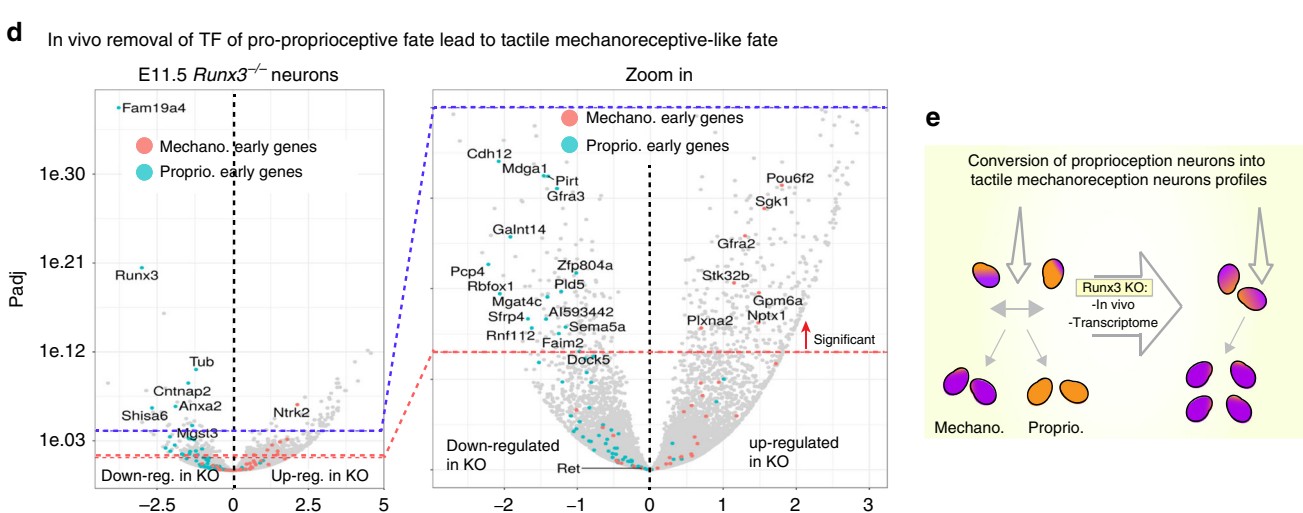

**Fig. 5 Early branch-specific gene *RUNX3* as a key player for fate biasing before split decision. a** Bifurcation analyzed. **b** Pseudotime of *Runx3* along the trajectories show that *Runx3* is upregulated in cells before the bifurcation point. Colors encode branches, as in Fig. 4a. **c** DRG sections from *Runx3+/+;Bax−/−* and *Runx3−/−;Bax−/−* E12.5 embryos (*n* = 4) stained for TRKB and ISL1 reveal a threefold increase in the number of TRKB+ neurons in the absence of *Runx3*. Scale bar, 50 µm. ***$P < 0.0001$. Data are presented as mean values ± SEM. **d** Volcano plots of differentially expressed genes between *Runx3+/GFP* and *Runx3GFP/GFP* DRG neurons (E11.5) showing downregulation of genes that belong to early genes modules of the proprioceptor trajectory and upregulation of genes that belong to the early genes modules of the mechanoreceptors trajectory. **e** Scheme recapitulating the findings.

GFP+ presumptive proprioceptors from E11.5 *Runx3-P2GFP/GFP* knock-in mice[30]. In those *Runx3* promoter-specific knock-in mice, GFP expression pattern faithfully reproduces RUNX3 expression during development. This mouse model can thus be used to trace and specifically analyze neurons of the RUNX3 lineage, expressing (in heterozygous state) or lacking (in homozygous state) *Runx3* expression. In GFP+ cells lacking *Runx3*, a large number of early and late genes identified in the pro-proprioceptor decision were significantly downregulated (i.e. *Runx3, Fam19a4, Cntnap2, Shisa6, Anxa2*, Fig. 5d, blue dots) while some identified as pro-mechanoreceptor were significantly upregulated (i.e. *Ntrk2, Gfra2, Pou6f2, Gpm6a, Plxna2*, Fig. 5d, red dots), suggesting the existence of competing genetic programs prior to cell fate choice during sensory neuron diversification. Interestingly, these differentiation programs not only activate genes specific of one cell fate but also repress gene programs of the alternative fate. Despite the observation that the loss of RUNX3 in the early myelinated lineage results in a shift of the cell identity which underlines the weight of a single factor in maintaining the integrity of a differentiation program, our computational analysis suggests a combinatorial coding of the cell identity, as previously suggested[27,31].

Together, those data strongly suggest an essential role of gene expression dynamic prior to and at the bifurcation point in unfolding the transcriptional programs that defines fate choice towards proprioceptor and mechanoreceptor fates and that competing programs between opposing fate determinants is an effective means to simultaneously promote one fate and exclude the other.

**Convergence of transcriptional programs during neuronal differentiation**. Our last investigation was to define the specific signature of the most differentiated clusters, the endpoints of the trajectories (Fig. 6a). Also, based on intersectional gene module of all endpoints (i.e. common genes), we ought to reveal the possible existence of a sensory neuron module that would define the sensory lineage and which would be passed-on by the mother cell (the cycling progenitors) to all the daughters cells (postmitotic sensory neurons). Our data show that although the differentiation trajectories exhibited divergent intermediate paths, they all seem to converge again at a transcriptional level as they mature to distinct neuron types, as judged by the relatively low number of transcriptional heterogeneity between neuron types at the end-points of our analysis compared to the progenitors population (Fig. 6b) and the high overlapping of GO term categories (Fig. 6c). Our analysis could identify around 120−180 differentially expressed genes between clusters constituting the endpoints of the trajectories in branches A and B (Source Data file and the

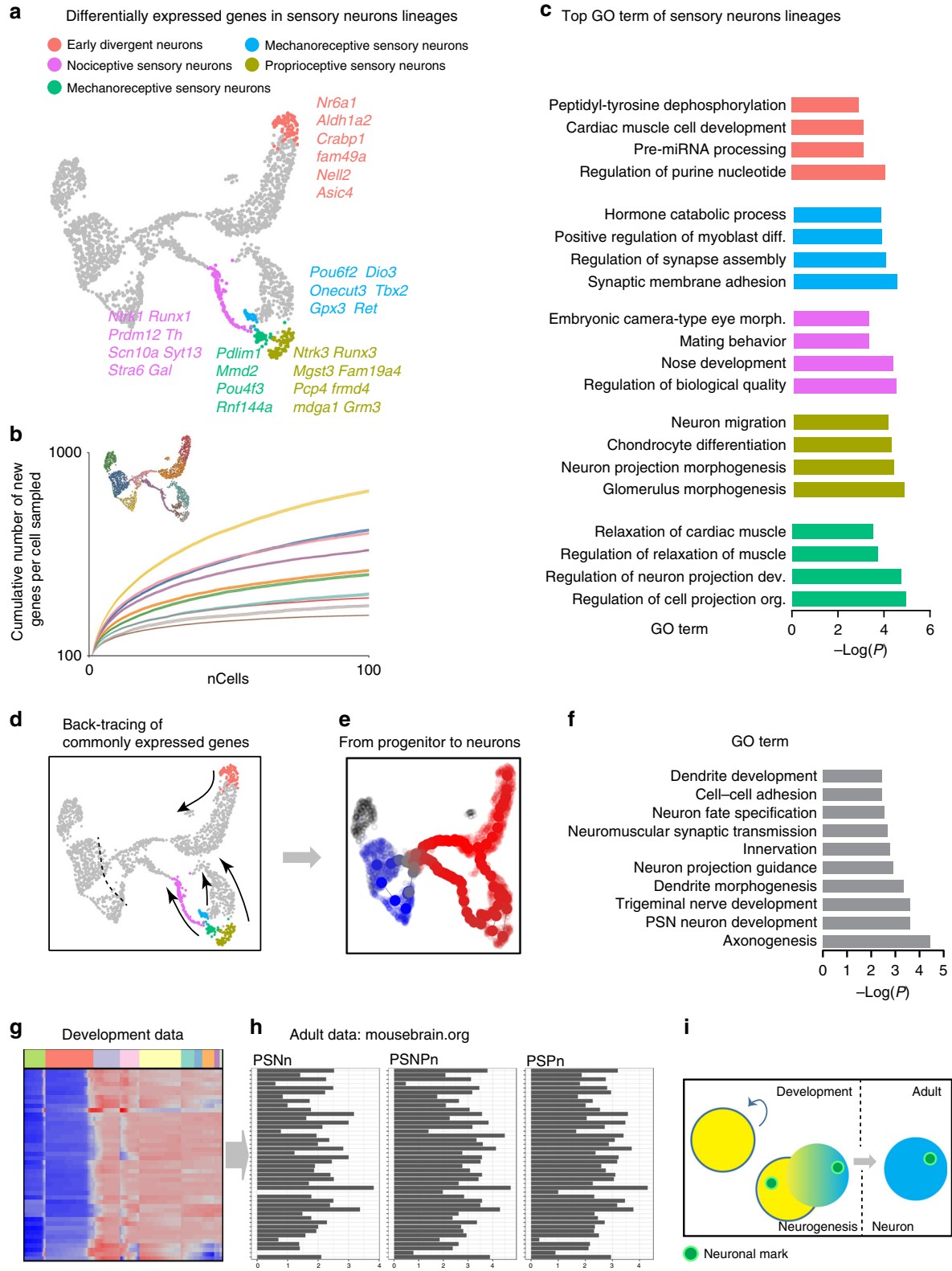

final recapitulating scheme (Fig. 7)). Interestingly, some of these genes are markers specific of subpopulations (Fig. 6a) identified later during development. For example, within the nociceptor lineage, *Trpv1*, *Th* and *Trpm8* were found to be expressed already at E12.5, yet they will each participate in the classification and function of discrete nociceptive neuron subtypes in adults[8], confirming the results of a recent study[26]. Altogether our findings

provide a general principle for diversification of sensory neurons into subtypes which involves repression of specific genes within intermediate cell states, leading to emergence of neuronal types with defined functional properties.

The convergence of the population at the transcriptomic level led us to examine whether common module genes were detectable at E12.5 and if they could be found at earlier

**Fig. 6 Transcriptional identity of the major subtypes at neuronal endpoints. a** Analysis of endpoints trajectories identifies differentially expressed genes in the distinct sensory lineages. **b** Bootstrapping analysis detects a decrease in the number of unique genes as the cells progress towards differentiated states along the diversification tree, with the endpoints clusters having the lowest heterogeneity among all clusters. **c** GO terms of the sensory lineages show redundancy of the categories between the subtypes of neurons. **d–f** Gene transmission from progenitor cell to daughter neurons was studied by first identification of the common genes at the trajectories' endpoints and second by back-tracking those modules of genes in time up to the cycling neuronal progenitors (**d**). We identified 43 genes being already expressed by the progenitor population; those genes belong to GO term categories in (**f**) and are all related to neuronal processes. **g, h** Pseudotime of the 43 identified genes during development (**g**, clusters color code is similar to Fig. 1c) and still expressed in the adult DRG neurons (**h**) composed of the three sub-taxons: peripheral sensory neurofilament neurons (PSNn), peripheral sensory non-peptidergic neurons (PSNPn) and peripheral sensory peptidergic neurons (PSPn) (data from mousebrain.org). Note one line in (**g**) and (**h**) represent one same gene. The genes list is provided in the Source Data file. **i** Scheme showing the ID neuronal mark pass on by mitotic mother cell to daughter neurons and kept into adulthood.

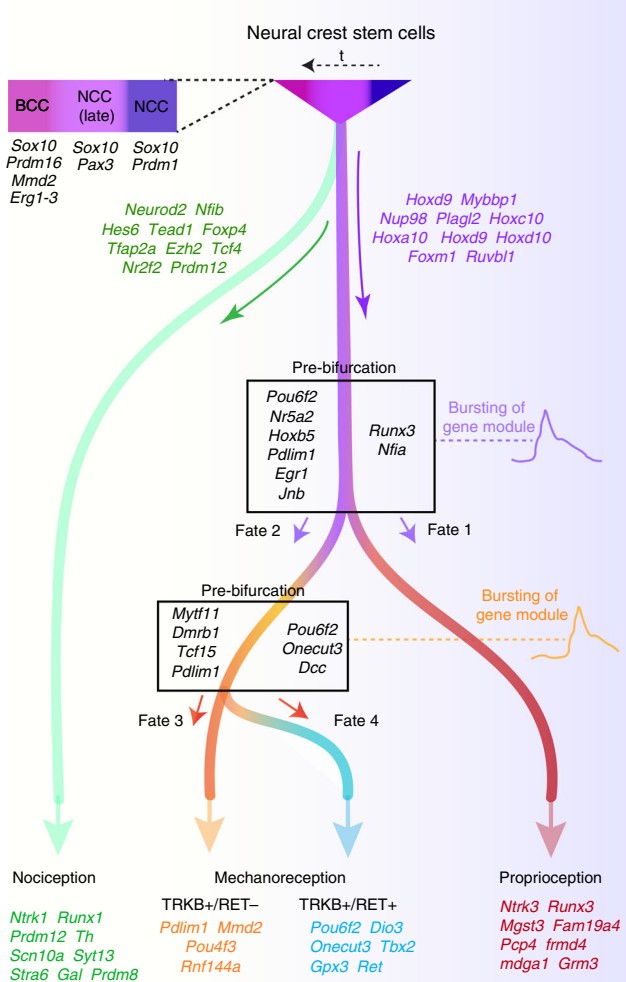

**Fig. 7 Schematic representation of the findings.** Schematic representation of the neurogenesis from different stages of NCCs via distinct trajectories for the neurons of the unmyelinated (nociceptive) versus myelinated (tactile mechanoreceptive and proprioceptive) lineages of our dataset. The genes predicted to be involved in the unfolding of the trajectories are represented at the starting point of the trajectories, while genes that define the latter stage of maturation in our dataset are represented on the endpoints of the trajectories. Note the distribution of some members of the PRDM family (putative histone methyltransferases) throughout the tree. From the trajectory leading to tactile mechanoreceptor and proprioceptive neurons, bursting of genes of competing genetic programs prior to binary fate decision during sensory neuron diversification is reported at the pre-bifurcation level. Those genetic programs direct towards either specific neuronal fate.

developmental stages and preserved throughout the life. To identify such a program that defines the entire sensory lineage, we back-traced modules of genes related to neuronal function expressed at E12.5 (Fig. 6d, e) and already present at the root of neurogenesis. We identified 44 genes as being transmitted from mitotic neuronal progenitors to all post-mitotic sensory neurons at E12.5, and which code for generic neuronal features that determine the neurotransmission phenotype and morphological features specific to neurons (most represented GO terms: axon neuron projection, myelin sheath and cell projection) (Fig. 6f and Source Data file). Importantly, those genes were maintained at all levels of the diversification tree and were still found in adult sensory neurons using dataset from Mousebrain.org[8] composed of three sub-taxon: peripheral sensory neurofilament neurons, peripheral sensory non-peptidergic neurons and peripheral sensory peptidergic neurons (Fig. 6g, h). Our data thus provide insight into a set of genes defining the entire sensory lineage from sensory progenitors to adult sensory neurons (Fig. 6i).

## Discussion

The molecular mechanisms regulating the fate of the somatosensory neurons are not entirely understood mainly because of the specific challenges including a relatively fast neurogenesis process and substantial mixing of progenitors and cell types at the location where the sensory ganglia coalesce. As a result, the current paradigm established only a fraction of molecular determinants essential for the development of particular sensory fates[5]. Moreover, these previously established molecular programs are likely associated with the processes of implementation of a fate rather than they are related to a process of a fate choice, which might include complex accumulation of cell history, activation of biasing factors and the integrative role of dynamic signaling landscape. In this study, we provide a comprehensive scRNAseq-based analysis of the neurogenesis in the mammalian somatosensory system and identified unique and mixed-lineage transcriptomic states that evolve to culminate in specific neuronal subtypes (Fig. 7). Moreover, our study provides an analysis of the developmental dynamic throughout early stages revealing the emergence of early somatosensory neurons. Our findings identify a combinatorial process through which cell type-specific neuronal identities emerge and reveal the preceding fate-related changes in heterogeneous population of early dividing progenitors and neurons prior to fate branching points in the diversification tree leading to concrete sensory subtypes.

Our branching model helps in understanding the hierarchy of the sensory neuron lineage and reveals sequential binary decisions yielding to the main sensory subtypes. Such binary fate decision models[32] are recognized for increasing the robustness and reproducibility of cell type distribution (for review, see ref. [33]). Binary decision among cycling cells leading to neuronal diversity has been largely described in the context of asymmetric division in progenitors or lineage pre-patterning of precursors earlier during development[1,33–36]. Our data indicate that newly born sensory

neurons within a trajectory derive from a relatively homogeneous population of cells and that molecular specificity in neurons is progressively acquired along the pseudo-time axis of differentiation by a progression of consecutive binary decision. The notion of branche segregation by mutual repressive activity had already been suggested for cell types diversification during late embryonic somatosensory development, for instance during the differentiation of TRKA[+]/RUNX1[+] nociceptors into a peptidergic and a non-peptidergic population of sensory neurons[5,37], yet the timing and gene-regulatory strategies that regulate this differentiation process remain poorly understood.

Our data suggest a mechanism of distributing the fates of early neuron types in the developing nervous system whereby fate-biasing heterogeneity of the post-mitotic neuronal population is gradually increasing until the commitment point in binary decision-making[38]. This is generally similar and conserved during the formation of other fates in the multipotent NCC lineage tree[32]. The molecular underpinning of the bifurcation events along the hierarchical decision tree of somatosensory development revealed initial co-activation of bi-potential transcriptional states in cells prior to bifurcation, followed by gradual shifts toward cell fate commitment and binary decisions. It suggests that determination of a specific sensory neuron fate during early development is therefore achieved by the increased synchronization of relevant programs (as key lineage priming factors) together with concurrent repression of competing fate programs. This complements the view of an early segregation of cell state and identification of cell fate based on differential but unique expression of genes or gene modules that are necessary for their neuronal differentiation[4,39,40]. While restricted neuronal expression (in an on−off pattern) based on progenitors' fate-limiting TFs have been largely demonstrated in the control of a binary fate choice[39,40], the sequence of transcriptional events and in general, the molecular mechanisms operating in neurons as they diversify have been difficult to appreciate. However, complex neuronal diversification programs might be considered as the collection of multiple binary fate decisions integrated over time. Therefore, the high-resolution analysis of the dynamic of transcriptional profiles in developing neurons presented here may thus provide an analytical basis for studying further how neuron diversity is generated in other parts of the nervous system.

Our analysis also complements a study recently published[26], which presented the transcriptional profile of the somatosensory neurons across various stages covering the embryonic and postnatal development. In contrast to our findings, Sharma et al. defined the early postmitotic sensory neurons as unspecialized, whereas in our study they are already differentiated into specific sensory sub-branches. Our data confirm previous results demonstrating neuronal heterogeneity in DRG as early as E11.5, with clearly defined populations of proprioceptor and tactile mechanoreceptor at this stage[13–17,28,41]. Overall, our studies mostly differ in their focus: Sharma et al. opted for analysis of large number of cells for offering a detailed developmental transcriptional atlas of sensory cell types while our study, focusing on fewer early time points, provides a mechanistic insight of the molecular events leading to the early emergence of sensory neuron diversity.

The mosaic differentiation pattern with a relatively constant proportion of generated sensory neurons and cell types within the somatosensory system makes it unlikely to be constrained by purely deterministic mechanisms. Instead, it raises the possibility of a stochastic fate decision within immature neurons in which one of their bivalent molecular programs would get stabilized and would thus progressively drive one of the two possible fates. Such mechanism has been demonstrated during the differentiation of photoreceptor neurons in *Drosophila*, where the stochastic induction of a TF, Spineless, controls cell fate decision cell-autonomously and simultaneously coordinates the alternate fate identity in neighboring photoreceptors[42]. In our system, we show that during decision between proprioceptor and mechanoreceptor lineages, high levels of RUNX3 expression are required to specify a proprioceptive fate. Hence, in *Runx3* mutants, cells of the RUNX3 lineage express molecular determinants characteristic of the early and late gene modules of the alternate tactile mechanoreceptive cell lineage. The mechanism by which *Runx3* becomes heterogeneously expressed in a subset of immature neurons remains however unknown and is most likely driven by differences in microenvironment including possible crosstalk with other differentiating subtypes or access to diffusing molecules. Upstream transcriptional regulators are likely functioning in coordinating the expression state of these key neuronal TFs. It will thus be interesting to see whether and how upstream mechanisms tightly control the activity of the various promoters and regulatory elements that promote *Runx3* expression specifically in the presumptive proprioceptor lineage[30]. Importantly, by integrating multiple steps of cell identity regulation, such mechanism operating along the hierarchical differentiation tree could help ensuring a correct representation of all neuronal subtypes developing from a common progenitor pool, especially via influencing the fate choice dynamics.

In conclusion, our work has provided a detailed transcriptional roadmap of neurogenesis and sensory neurons development that captured the dynamic of molecular events participating in cell fate choice, with progression through the bifurcation and lineage specification of the sensory neurons. We identified dynamic burst of modules of genes, which led to branching point within lineages (Fig. 7). The identification of genes modules that prime sensory neurons to specific fate might be of benefit for engineering-induced pluripotent stem cell and embryonic stem cell technologies, and help advancing our ability to engineer specific neuronal populations for basic research, tissue regeneration, or for screening drugs against neurodegenerative or pain disorders. Also, this might help deciphering the combinatorial code of TFs responsible for cell state and to manipulate it in a context of a loss of cellular homeostasis with the hope to reset the cells to a healthy state.

## Methods

**Animals**. Wild-type C57BL6 mice were used unless specified otherwise. *Plp1^CreERT2^*, *Wnt1^cre^*, *Isl1^Cre^*, *Ntrk3^Cre^*, *Runx3^−/−^*, *Fam19a4^YFP^*, *Neurog2^CreERT2^*, *Neurog1^Cre^*, *Bax^−/−^* and *Ai14 (Rosa26^tdTOM^)* mouse strains have been described elsewhere[14,25,28,43,44]. Animals of either sex were included in this study. Animals were group-housed, with food and water ad libitum, under 12 h light–dark cycle conditions. All animal work was performed in accordance with the national guidelines and approved by the local ethics committee of Stockholm, Stockholms Norra djurförsöksetiska nämnd.

**Treatment**. For cell fate tracing experiments with inducible mouse lines, tamoxifen (Sigma, T5648) was dissolved in corn oil (Sigma, 8267), and delivered via intraperitoneal (i.p.) injection to pregnant females at E9.5 (100 μg/g of bodyweight).

For cell cycle study, intraperitoneal injection of pregnant females with EdU (100 mg/kg, Invitrogen) was performed at E10 and E12 days of gestation. Injected females were killed 2 h after injection for analysis. EdU incorporation was subsequently resolved using Alexa Fluor 488 azide according to the manufacturer's instructions (Invitrogen).

**Immunostainings and RNAscope® in situ hybridization**. Animals were collected, decapitated and fixed for 1–4 h at +4 °C with 4% paraformaldehyde in phosphate-buffered saline (PBS) depending on the stage, washed in PBS, cryopreserved in 30% sucrose in PBS, embedded in optimal cutting temperature (Tissue-Tek) and cryo-sectioned at 14 μm. In situ hybridization was performed using standard RNAscope protocol (ACDBio). The RNAscope probes used in this study are Mm-Fxyd7, Mm-Neurog1, Mm-Neurog2, Mm-Spp1, Mm-Ahr, Mm-Girk1, Mm-Cbln4, Mm-Chodl and Mm-Anxa2 (ACDBio). Sections were incubated for 24 h at +4 °C with primary antibodies diluted in blocking solution (2% donkey serum, 0.0125% NaN₃, 0.5% Triton X-100 in PBS). Primary antibodies used were: rabbit anti-RUNX3

(gift from T. M. Jessell), rabbit anti-RUNX1[44], goat anti-TRKA (1:400, R&D Systems AF1056), goat anti-TRKB (1:500, R&D Systems AF1494), rabbit anti-TRKC (1:1000, Cell Signaling 3376), mouse anti-ISL1 (1:250, Developmental Studies Hybridoma Bank 39.4D5), goat anti-RET (1:100, R&D Systems AF482), goat anti-TRKC (1:500, R&D Systems AF1404), mouse anti-βIII-tubulin (1:1000, Promega G712A), chicken anti-RFP (1:250, Rockland 600-901-379S), mouse anti-NF200 (Sigma, N0142; 1:500), mouse anti-SOX10 (1:1000, Santa-Cruz, #sc-36569). After washing with PBS, Alexa Fluor secondary antibodies (Live Technology; 1:500 in blocking solution) were applied overnight (at +4 °C). Samples were then washed in PBS and mounted in DAKO fluorescent mounting medium. Staining was documented by confocal microscopy (Zeiss LSM700) using identical settings between control and experimental images. Optical sections were 2 μm in ×20 overview pictures unless specified.

**Quantifications**. For cell type quantifications, ImageJ software was used. Only neurons with a visible nucleus were considered for analysis. Quantification of molecular markers in the DRG was carried out on five DRG sections/animal, selected from the most equatorial region of each DRG and covering the segments C5-T1 (see figure legends for *n* values and genotypes).

**Statistical analysis**. Data were analyzed using GraphPad Prism 6 and expressed as mean ± s.e.m. The statistical test performed is reported in the figure legend. *t* tests were two-sided. Legend for significance: *$P \le 0.05$, **$P \le 0.01$, ***$P \le 0.001$. No animals or data points were excluded from the analysis. Our sample sizes are similar to those generally employed in the field.

**Single-cell isolation for single-cell analysis**. Brachial DRGs were dissected (from E11 embryos) or whole embryonic trunks collected (E10.5 and younger) in Leibovit'z L-15 medium (Life Technologies) on ice. The tissue was incubated in 0.05% trypsin-EDTA (1×) (Life Technologies) for 5, 7, 10 and 15 min for tissue obtained from E9.5, E10.5, E11.5 and E12.5, respectively, at 37 °C, in a thermomixer comfort (Eppendorf) at 700 RPM. After spinning down the samples at 100 RCF for 5 min, the supernatant was removed and replaced by Leibovit'z L-15 medium (Life Technologies). The tissue following enzymatic digestion was physically triturated using two different sizes of pipettes previously coated with 0.2% bovine serum albumin until the solution homogenized. The cell suspension was then filtered through a 70-μm cell strainer (BD Biosciences) to remove the clusters of cells.

**FACS**. For the scope of this work, we sorted TOMATO+ cells from different transgenic lines (both inducible and non-inducible) using either instruments from BD (manufacturer) (BD FACSDiva 8.0.2 for *Wnt1cre;Ai14*, *Plp1creERT2;Ai14* and *Ai14* and BD Influx for *Isl1cre;Ai14* and *Ntrk3cre;Ai14*. The software used is FlowJo v10 (BD) and BD FACS Software 1.2.0/142/Utopex 1.2.0.108 respectively. The gating strategy is shown in Supplementary Fig. 2. Debris and erythrocytes were gated out from the total events using FSC-A versus SSC-A plotting. Doublets were further gated out using FSC-A versus FSC-H plotting. Finally, we plotted FSC-A versus PE-A (corresponding to TOMATO+ signal) to select cells that were TOMATO+ (Supplementary Fig. 2a, b, e) or we plotted the FSC versus the 585/29 [561]-tdtomato (corresponding to TOMATO+ signal, Supplmentary Fig. 2c, d). A cell preparation negative for TOMATO was used to define the negative population (Supplementary Fig. 2e). Note, the BD Influx instrument was manually set up and calibrated on daily basis using BD CST and BD FACS Accudrop beads. Single TOMATO+ cells were sorted by fluorescence-activated cell sorting (FACS) into individual wells containing lysis buffer in 384-well plates provided by the facility (Karolinska Insitutet). All plates were immediately placed on dry ice and stored at −80 °C before processed for Smart-seq2 protocol at the single-cell facility (Karolinska Institutet).

**Single-cell sequencing and generation of count matrices, QC and filtering**. The single-cell transcriptome data were generated, using Smart-seq2 protocol at the Eukaryotic Single-cell Genomics facility at Science for Life Laboratory in Stockholm, Sweden. This sequencing approach does not use UMI and hence does not correct for PCR duplicates. The samples were analyzed by first demultiplexing the fastq files using deindexer (https://github.com/ws6/deindexer) using the nextera index adapters and the 384-well layout. Individual fastq files were then mapped to mm10_ERCC genome using the STAR aligner using 2-pass alignment[45]. Reads were filtered for only uniquely mapped and were saved in BAM file format, count matrices were subsequently produced. Estimated count matrices were gathered and converted to a SingleCellExperiment object, QC metrics were computed using calculateQCMetrics function. Cells having less that $5 \times 10^4$ transcripts, less than 2500 genes and more than 25% of proportion of ERCC reads were filtered out.

For the selection for the developing sensory cells, the main analysis of the count matrix was mainly performed using pagoda2 R package[12]. The filtered overview count matrix gene expression variance was adjusted (pagoda2, $k = 10$) and 5801 overdispersed genes were detected. PCA was performed on the overdispersed genes (pagoda2, nPcs = 100, maxit = 1000) and 13 PC were retained using elbow curve selection. KNN graph (pagoda2, $k = 40$, centered, cosine distance) and UMAP visualization (umap python, $n\_neighbors = 30$, min_dist = 0.5) were generated in PCA space. Clusters were then identified on the KNN graph using leiden

algorithm[46], and pathway overdispersion analysis was performed (pagoda2, correlation.distance.threshold = 0.95) to identify relevant biological aspects. Differential gene expression was performed for each of the detected clusters using Wilcoxon rank test. Cells in clusters with *Sox10* gene positively differentially expressed were kept, as well as linked neurogenic clusters (using aspect1).

To cluster and visualize the developing sensory data, the subsetted count matrix was further processed using pagoda2 with similar parameters as for the overview (4887 detected overdispersed genes), clusters were detected using infomap algorithm in addition to leiden for a finer clustering. A biological aspect linked to mitochondrial respiratory chain has been identified to be different among batches from the same condition. This aspect was regressed on the raw count matrix using ScaleData from Seurat package[47], the regressed count matrix was further processed through the Seurat 2 pipeline, variable genes identification (default parameters) and PCA (pcs.compute = 100), eight PCs were retained using elbow curve selection. UMAP was performed on the retained PCs ($n\_neighbors = 100$, min_dist = 0.5).

For the heterogeneity analysis, a bootstrapping analysis was performed for each timepoint separately to validate that cell type heterogeneity decreases upon sensory neuron differentiation. One hundred cells were randomly one after the other sampled with replacement, top 100 most expressed genes per selected cell were identified and the number of newly identified genes were added to the number of unique genes seen before. This process was repeated 100 times for each timepoint.

Count matrix correction was performed prior to pseudotime analysis, the raw count matrix was corrected using scde R package. scde fits error models for each cell in order to estimate the drop-out and amplification biases on gene expression magnitude[48].

In order to infer trajectories on the resulting data, diffusion maps were first computed using Palantir[49] python package (run_diffusion_maps, default parameters) on PCA space. Principal graphs were fitted on the first five diffusion components using ElPiGraph[50]. To avoid the E12.5 clusters describing the proprioceptors and the mechanoreceptors to be wrongly associated with the E12.5 nociceptive lineage, the principal tree was performed in several steps. First, a principal tree of 30 nodes was generated on the E10.5/11.5 subset of the data, by excluding leiden clusters 5, 6, 7, 9, 10, and infomap cluster 23. Proprioceptor cluster (leiden 10) was added by linking its geometrical center in diffusion space to the tree. Mechanoreceptors cluster (leiden 9) was added by inferring a 4-node principal tree in diffusion space on this subset, and by linking it to the E10.5/E11.5 tree using closest node. Trajectory for nociceptive lineage was recovered by inferring a 15-node principal curve in diffusion space on leiden clusters 5 and 6, the resulting principal curve was then linked to the whole tree by using closest nodes. In order to have branches that have at least more than one node, additional nodes were added by separating each edge of the tree in two equidistant ones.

Downstream analysis was performed using a slightly modified crestree R package (https://github.com/LouisFaure/crestree) provided by the previous neural crest study[32], in which precise description of the underlying statistics is mentioned. The difference in its usage is first the inference of a principal tree with ElPiGraph, which contains two roots and is mapped only once, and second the usage of the updated JASPAR2018 database[50] for TF activity inference. Associated genes with the tree were detected using test.associated.genes (default parameters) on the log transformed gene expression. Associated genes were fitted using fit.associated.genes (gamma = 5). The fitted profiles were clustered into 30 major patterns with hierarchical clustering using Ward method and Euclidean distance. Inference of TF activity was performed on 101 selected TF using activity.lasso (default parameters).

Bifurcation analysis was performed in three steps, genes differentially upregulated after bifurcation point were detected using test.fork.genes (default parameters), and differentially expressed genes were then assigned between two post-bifurcation branches using branch.specific.genes. Optimum expression and time of activation were estimated for each of the detected gene using activation.fork (default parameters), allowing the separation between early and late modules by setting a pseudotime threshold before the bifurcation. To analyze molecular mechanisms of cell fate selection, cell composition was approximated by a sliding window of cells along the pseudotime axis, cells were manually selected in order to represent the different steps of differentiation. The local gene−gene correlation reflecting the coordination of genes around a given pseudotime *t* was defined as a gene−gene Pearson correlation within each window of cells. The local correlation of a gene *g* with a module was assessed as a mean local correlation of that gene with the other genes comprising the module. Similarly, intra-module and inter-module correlations were taken to be the mean local gene−gene correlations of all possible gene pairs inside one module, or between the two modules, respectively[51].

For RNA velocity analysis, BAM files from each plate were processed using python command-line velocity tool[10], using run-smartseq2 command with GENCODE M21 genome and repeat masker annotation files, leading to a loom file for each plate containing spliced and unspliced transcript counts, which are then combined in one loom dataset, and cells are filtered out according to the cells kept in the final developing sensory dataset. Using scvelo python tool[52], genes having <20 spliced counts or genes having <10 unspliced counts were excluded and the 4000 top highly variable genes were kept. In addition, cell cycle genes are also filtered out from the analysis. PCA was performed on the spliced matrix, keeping 30 principal components and kNN neighbor graph was produced with $k = 30$. Moments of spliced/unspliced abundances, velocity vectors and velocity graph were

computed using default parameters. Extrapolated states are then projected on the UMAP embedding produced during the initial analysis step.

In order to compare the two neurogenic waves, the early E10.5 and late E11.5/12.5 neurogenic waves, associated genes with each waves were identified using test. associated.genes (default parameters) on their respective subtrees, having as a starting point the first crossing point of the tree, and the endpoint the next bifurcation for the early wave, or the endpoint of the late wave. Common genes were identified by the intersection of the two set of detected genes, the other genes were considered as wave specific. For common and differing genes respectively, early and late genes were identified separating the fitted profiles in two clusters via hierarchical clustering using Ward method and Euclidean distance.

Endpoints trajectories identity were defined by performing differential gene expression using Wilcoxon rank test on infomap clusters 8, 10, 15, 20 and 22, genes with log2fc higher than 1 and expressed in more than 90% of the cells of a given cluster were considered as a marker for identity. GO term enrichment analysis was performed with topGo R package on these markers for ontologies "biological process" as well as "cellular compartment". Test was performed using "elim" algorithm and "fisher" statistical testing.

For the comparison with Runx3 KO bulk dataset, differential gene expression results of bulk data from Apple et al.[30] were obtained from GEO database (GSE81140), early genes for each bifurcation were overlaid on the volcano plot to identify their distribution of the genes between wild type and Runx3 mutants.

To identify the common early neuronal identity, the tree was separated in two subtrajectories, with one having root in the E10.5 NCCs population, and endpoints E10.5/E11.5 and E12.5 mechanoreceptors and proprioceptors, and the other having the E12.5 NCCs population as root and the E12.5 nociceptive neurons as endpoint. A gene defining neuronal identity is defined as a gene being activated in the NCCs pool (pseudotime < 6) and present in more than 98% of the cells which at pseudotime > 6 in all each single linear trajectory (from root to end), activation. statistics was used to identify early genes being activated in the NCCs pool. Expression of each of the detected gene markers was checked in adult mice, by looking at their levels in aggregated cluster data of peripheral sensory neurons from mouse brain atlas[8] (http://mousebrain.org/loomfiles_level_L6.html).

**Reporting summary**. Further information on research design is available in the Nature Research Reporting Summary linked to this article.

## Data availability
The authors declare that all data supporting the findings of this study are available within the article and its supplementary information files or from the corresponding author upon reasonable request. Raw sequencing data have been deposited in the GEO database under accession code: GSE150150. This includes a pagoda2 web file (p2w_sensory.bin) allowing exploration dataset on a web browser. The file can be opened on the following link: http://pklab.med.harvard.edu/nikolas/pagoda2/frontend/current/pagodaLocal/index.html. For the Comparison with Runx3 KO bulk dataset, differential gene expression results of bulk data[30] were obtained from the GEO database (GSE81140). Source data are provided with this paper.

## Code availability
All codes for scRNAseq data preprocessing and pseudotime analysis are deposited under the form of notebooks on the following github repository: https://github.com/LouisFaure/sensoryfates_paper. Source data are provided with this paper.

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

## Acknowledgements
We thank Y. Groner and D. Levanon for the *Runx3* mice; C. Ibanez for the *R26^tdTOM* mice; A. Moqrich for the *Fam19a4^YFP*; T. Jessell for the RUNX3 antibody; the CLICK imaging Facility supported by the Knut and Alice Wallenberg Foundation; Single Cell Facility at SciLife Laboratory. We are thankful to Jaromir Mikes for sorting the cells. We are grateful to Prof. Patrik Ernfors and Prof. Abdel El Manira for critical reading of the manuscript. We are also grateful to Dr. Ruslan A. Soldatov (Kharchenko Lab) for sharing codes. I.A. was supported by ERC Consolidator grant "STEMMING-FROM-NERVE", Bertil Hallsten Foundation, Swedish Research Council, Paradifference Foundation. F.L. was supported by the Swedish Research Council (VR), the Ragnar Söderberg Foundation, Knut and Alice Wallenberg Foundation, Swedish Brain Foundation, Karolinska Institutet, the Karolinska Institutet Strategic Research program in Neuroscience (StratNeuro) and the Ming Wai Lau Foundation. L.F. is supported by the Austrian Science Fund DOC 33-B27. S.H. is supported by the Swedish Research Council (VR), the Swedish Brain Foundation and Karolinska Institutet. Open access funding provided by Karolinska Institute.

## Author contributions
Study design and supervision: S.H. Tissue collection and acquisition of data: Y.W., M.E.K., P.F., K.K.Y.C, C.P., H.W., L.L.S., K.R., L.C., M.A.L., H.C.L., A.d.C. and S.H. Analysis and interpretation of data: L.F. and S.H., with inputs from M.E.K., F.L. and I.A. Figures: S.H. Drafting of manuscript: L.F., F.L., I.A. and S.H. with inputs from all co-authors.

## Competing interests
The authors declare no competing interests.
