## [Peer Review File · Nature Communications]

Reviewers' Comments:

Reviewer #1:

Remarks to the Author:

In this study, Faure et al present an interesting data set to analyze transcriptional changes that occur in the somatosensory lineage during early development. Specifically, the author perform scRNA-seq analysis on neural crest cells to post-mitotic neurons in the time between E9.5-E12.5. By monitoring transcriptomic changes in this age range, the authors propose the model that sensory neuron diversity is achieved by a transition through an intermediate state characterized by co-expression of genes present in multiple cell classes. The authors support this model primarily through a well-executed set of computationally inspired descriptive analyses, including RNA-velocity and pseudotime analysis. The authors also include some data from the Bax;Runx3 compound mutant mouse to support the idea that co-expressed gene modules 'compete' with each other during development to establish cell fate.

Overall, I feel the data quality, particularly of the scRNA-seq, is of very high quality as is the analysis of these data. I have several questions that I feel should be address prior to publication, particularly with regards to the wording behind some of the conclusions drawn in the text.

1. The authors appropriate cite the historical literature indicating there are multiple waves of neurogenesis in the somatosensory system. However, the fate mapping experiments with Neurog1-CreERT2 and Neurog2-CreERT2 (E9/10 TAM) show labeling of both large (wave 1) and small (wave 2/3) diameter neurons. This seems to argue that there is effectively only one wave of neurogenesis. Can the authors discuss this in the revision?

2. The Bax;Runx3 compound mutant is certainly an interesting experiment and really represents the only manipulation presented in this study. The authors show that some mechanoreceptor associated genes become upregulated in the Runx3-KO and that there is an increased number of TrkB-positive cells in the ganglia at E12.5. These data are used to support the conclusion that mechanoreceptor fates must be suppressed to lead to proprioceptors. I believe the caveats of this experiment should be stated more clearly to better allow the reader to grasp the findings. For example, the data described in line 311-313 is immunostaining data for one gene, although it is certainly an important gene, it is not necessarily proof that these cells have converted to a mechanosensitive fate. Furthermore, the data described the rest of Figure 5 is from bulk-RNA-seq, therefore a caveat here is that we do not know with confidence the identity of the cells in which the reported genes are changing.

3. Given that the last timepoint described in this study is E12.5, a timepoint where small diameter diversification is not yet seen, I think it would be useful in the abstract and elsewhere to mention that the conclusions of the study may only apply to early fate decisions (mechanosensory vs nociceptor) instead of 'final sensory subtypes (line 34)'

4. In line 390-393 the authors suggest this is the "...first scRNA-seq based analysis of the full course of neurogenesis in the mammalian somatosensory system...". I think this is a bit of an overstatement, and really not necessary, as Soldatov et al (citation 23) has analyzed a similar age range as well.

Reviewer #2:

Remarks to the Author:

In Faure et al., the authors seek to use single-cell transcriptomics to understand the gene expression landscape that determines neuronal cell fate decisions. They focused on early somatosensory neurons as these neurons have been extensively characterized and populations defined, but only key transcriptional components have been identified. Much is unknown about

the internal dialogue between cell fate programs that precedes neuronal type decisions. Faure et al. find that indeed competing transcriptional programs are active in maturing neurons that correspond to, in the cases examined here, binary cell fate decisions. This work is important in that it provides extensive context for the key transcriptional components leading to fate splits and identified new (and salient) subtleties these decision points (e.g. Neurod1/2 transient coexpression in Branch B). With revision, this manuscript should be considered for publication.

Major

1. The UMAP of "sensory neuron lineage" is a conglomerate of DRG cells from different ages and different genotypes. While this is a clever approach to target the key neuronal progenitors and populations, how well does this method recapitulate normal lineage? To increase my confidence that this piecemeal approach is representative of normal sensory neuron lineage, I would like to see a side-by-side of this cohort of transcriptomics from multiple lines with a small run of wild-type DRG cells that are not genetically labeled or an explanation in the text why the authors feel this is unwarranted.

2. The authors' use of the Runx3^{-/-};Bax^{-/-} line was good to support their idea branched paths. However, much of this work has been done previously (Levanon et al., 2002; Inoue et al., 2007; Souichiro et al., 2008). Further, this work shows that a branch point exists but does not require a broad program as the authors conclude. I would like to see an experiment where cultured DRG neurons have genes previously not known to be involved before this study as being involved in pre-bifurcation "prepping" for fate decision be misexpressed, both up and down, with CRISPR or other genetic tools to show that these programs do act as a switch in these fate decisions.

Minor

1. Not spending a chunk of the discussion contrasting these findings with the new Ginty paper (Sharma et al, 2020) seems like a fairly glaring omission.

2. I can't find how many cells were used. Please state the total number of cells and how many were used for each genotype and each age.

3. Claims about cell fate bifurcations and lineage seem a bit strong given that much of the information is inferred. Since the data is pooled between FAC-sorted subpopulations from different genetic lines, it is impossible to know if these splits into different populations are artifacts of the single cells that were enriched for or indicate real lineages. The reporter lines were carefully thought out, but as the authors themselves show, as with the NeuroD1/2 coexpression, there is much we still do not know.

4. While this work does an excellent job proposing that cell fate decisions are antagonizing programs rather than a few key genes, the proposed model in figure 7 is remarkably similar to the previous model for the somatosensory system (Lallemend and Ernfors, 2012). The idea of sequential binary branches for somatosensory development is well established.

5. Quantification of the colocalization in fig1i might help to make the point about choice points better

6. It might be nice to somehow incorporate information about migratory waves of NCC into your high dimensional analysis. For example, what happens to the map if you subtract boundary cap cells?

7. It's not entirely intuitive what fig 1j is showing. Could be more explicit about what the colors mean in the legend or better yet in the panel itself.

8. The paper is logically laid out but I think this could use one more pass at line editing. "that remodeling" is duplicated on line 271

Reviewer #3:

Remarks to the Author:

I generally like single-cell datasets, as they provide a very nice unbiased view on a particular biological process. I do have a few concerns regarding this particular manuscript though:

1) I wish the authors would make a bit more of an effort to share their data with the community. To only deposit raw data on GEO is rather unusual these days. I would have liked for them to at least put up some Excel data for people to download or ideally a file that can be visualised in browsers like loompy or Loupe (though that may be difficult given that this is not 10X).

2) Related to this: if they want their article to be cited and appreciated by regular sensory/developmental biologists, they may want to spend a bit more time explaining how the various algorithms they use work conceptually. How does RNA velocity “reveal a clear directionality” for example? Does it use the temporal nature of the dataset to cluster cells that look similar over time?

3) Finally, while I appreciate that we are all required to oversell our data slightly to get into journals such as these, I fear the authors may not convince many readers that they have found anything particularly new. This is a point that I don't think requires any action, but just to express my feeling that when I read this, I was trying hard to figure out how exactly this was very different from what has been reported before? Apart maybe from the Neurog1/2 finding (see below). Personally, I do not think this should prevent publication – on the contrary, I regret that we are not permitted to openly celebrate replication and concordance with prior literature in our papers.

Action points:

- Please indicate how you will improve data sharing
- Alter your text to explain a bit more about the algorithms you are using and how they work conceptually
- Add your FACS gating strategy to the supplementary and add details in the methods as to how you FACSeD (e.g. did you use a live/dead stain? What did you sort into, etc)
- Explain exactly how you batch controlled these experiments – this is of course of particular relevance given that you were examining differential expression over time.
- State somewhere clearly that you didn't use UMIs and how this affects data interpretation (i.e. you may have PCR duplicates in your data?)
- Provide some kind of quantification for Figures 2e&f. Right now, I'm not overly convinced.

Response to Reviewers' comments:

We are very glad that the reviewers find our study of importance for the field, and of very high quality. We would like to thank them also for their constructive comments which we think have helped us improving our study. Please find below our answers to each point raised, in *italic* is the text that have been inserted in the manuscript according to reviewer comments.

Reviewer #1 (Remarks to the Author):

Overall, I feel the data quality, particularly of the scRNA-seq, is of very high quality as is the analysis of these data. I have several questions that I feel should be address prior to publication, particularly with regards to the wording behind some of the conclusions drawn in the text.

1. The authors appropriate cite the historical literature indicating there are multiple waves of neurogenesis in the somatosensory system. However, the fate mapping experiments with *Neurog1-CreERT2* and *Neurog2-CreERT2* (E9/10 TAM) show labeling of both large (wave 1) and small (wave 2/3) diameter neurons. This seems to argue that there is effectively only one wave of neurogenesis. Can the authors discuss this in the revision?

We understand the concern of the reviewer, who is referring to the 2 neurogenesis waves based on their timing of cell cycle exit and of their differential expression of *NGN2* and *NGN1* (Carr and Simpson, 1978; Lawson and Biscoe, 1979; Ma *et al.*, 1999). In this context, our results indicate that a discrete expression of the two NGNs would not predict cell fate commitment to either myelinated or unmyelinated lineage, at least for a large proportion of sensory neurons. While at odd with previous assumptions, a recent study by Sharma *et al.* (Nature, 2020) – using a similar Cre-based strategy and single cell sequencing – also demonstrates that the waves of neurogenesis are not linked to the single expression of *NGN1* for example. This is supported by both our single cell RNAseq data and our Cre-based lineage tracing. This apparent discrepancy between the recent studies (Sharma *et al.* and ours) and previous work (Ma *et al.*, 1999) does not lie within the conclusion that *NGN2* and *NGN1* would functionally define broadly the 2 waves, but in the fact that this function of the NGNs would be linked to their strict mutually exclusive expression. Our data show early induction of *Neurog2* followed by that of *Neurog1* in the same progenitor cells, with the possibility that at the very beginning and end of the neurogenesis waves, co-expression of *Neurog2* and *Neurog1* might not be found. This supports the previous conclusion that the specific expression of NGNs would not induce a particular lineage (Ma *et al.*, 1999; Lallemand and Ernfors, 2012). Yet, our data suggest that the relative levels of the two NGNs across time (and certainly expression of associated genes and/or the chromatin state of the progenitors) would define the waves, explaining why the loss of *Neurog1* leads to a loss of the second wave (Ma *et al.*, 1999). We think therefore that there are two waves of neurogenesis, as previously described, but that a restricted expression of *NGN2* and *NGN1* does not define them. Our results rather complement previous observations. We thank the reviewer to bringing this concern to us as these results could be misleading. We have now added a sentence summarizing this interpretation of our findings in regard to previous literature, and which we believe are not in contradiction with 2 sequential neurogenesis waves, but instead ponder the requirement of either neurogenins to be timely defined since most neuronal progenitor cells sequentially express the two neurogenins.

The following text has been included in the manuscript to clarify this (results section, paragraph Neural crest stem cell populations and waves of sensory neurogenesis, line 222 to line 228):

“Hence, we conclude that most sensory neuron progenitors sequentially express Neurog2 then Neurog1 with co-expression in an intermediate state where both genes are co-expressed and suggest that the neuronal precursors generating the first or the second waves of neurogenesis are timely poised to preferentially require either NGN2 or NGN1 for neuronal differentiation (Fig. 2)”.

2. The Bax;Runx3 compound mutant is certainly an interesting experiment and really represents the only manipulation presented in this study. The authors show that some mechanoreceptor associated genes become upregulated in the Runx3-KO and that there is an increased number of TrkB-positive cells in the ganglia at E12.5. These data are used to support the conclusion that mechanoreceptor fates must be suppressed to lead to proprioceptors. I believe the caveats of this experiment should be stated more clearly to better allow the reader to grasp the findings. For example, the data described in line 311-313 is immunostaining data for one gene, although it is certainly an important gene, it is not necessarily proof that these cells have converted to a mechanosensitive fate. Furthermore, the data described the rest of Figure 5 is from bulk-RNA-seq, therefore a caveat here is that we do not know with confidence the identity of the cells in which the reported genes are changing.

We agree with the reviewer that we should state more clearly the limitation of the data described in line 311-313 of the original manuscript. This has been added in the text in Results section, line 333 to line 341: *“we observed a three-fold increase in the proportion of TRKB⁺ cells in E12.5 DRG (Fig. 5b,c), similar to previous results (Kramer et al.2006). This suggests a fate change toward mechanoreceptor fate in the absence of RUNX3, yet, cannot conclude on a shift of identity. Therefore, to investigate further a potential cell fate change ...”*

Regarding the experiment described in the rest of the Fig. 5, on the comparison of sequencing data from *Runx3* null cells, we apologize for not being clear enough in the original version of the manuscript and thank the reviewer for giving us the opportunity to clarify the findings. We have now added a new text describing the associated results, and we hope have answered the reviewer’s comment. Indeed, while we are using the sequencing dataset from *Runx3* null cells (from a *Runx3* knockin mice where GFP replaces *Runx3* allele), it is not a bulk-seq of the whole DRG, but only of the cells that would have been RUNX3⁺ in a WT condition and would constitute the presumptive proprioceptors (Appel *et al.*, 2016). These presumptive proprioceptors were FAC sorted based on the activity of the *Runx3* promoter, with presumptive proprioceptors were fluorescence activate lacking *Runx3* allele (*Runx3*^{GFP/GFP}) in one condition, and the other condition representing the presumptive proprioceptors expressing *Runx3*, i.e. *Runx3*^{+GFP}. Hence, this strategy targets exclusively the cells with active *Runx3* promoter activity among all cells of the DRG (please see details below).

In order to clarify this, we have added the following changes in the manuscript in the Results section, line 341 to line 349:

“Therefore, to investigate further a potential cell fate change, we mapped the genes composing the early modules that we previously identified within the proprioceptor trajectory onto a published differential gene expression analysis of (FACS isolated and RNA sequenced) Runx3 null and Runx3+/- GFP+ presumptive proprioceptors cells from E11.5 Runx3-

P2GFP/GFP knock-in mice. In this Runx3 promoter-specific knock-in mice, a GFP construct is inserted within the P2 promoter region of Runx3, resulting in a Runx3 knockout model (GFP::GFP) and heterozygous cells (Runx3::GFP) where GFP expression pattern faithfully reproduces RUNX3 expression during development. This mouse model can thus be used to trace and specifically analyze neurons of the RUNX3 lineage, expressing (in heterozygous state) or lacking (in homozygous state) Runx3 expression. In GFP+ cells lacking Runx3, ...”

3. Given that the last timepoint described in this study is E12.5, a timepoint where small diameter diversification is not yet seen, I think it would be useful in the abstract and elsewhere to mention that the conclusions of the study may only apply to early fate decisions (mechanosensory vs nociceptor) instead of ‘final sensory subtypes (line 34)’

We agree with the reviewer and have now changed the text across the whole manuscript accordingly.

4. In line 390-393 the authors suggest this is the “...first scRNA-seq based analysis of the full course of neurogenesis in the mammalian somatosensory system...”. I think this is a bit of an overstatement, and really not necessary, as Soldatov et al (citation 23) has analyzed a similar age range as well.

We agree with the reviewer and changed the sentence accordingly, Discussion section, line 426 to line 429: “*In this study, we provide a comprehensive scRNAseq-based analysis of the neurogenesis in the mammalian somatosensory system*”

--

Reviewer #2 (Remarks to the Author):

In Faure et al., the authors seek to use single-cell transcriptomics to understand the gene expression landscape that determines neuronal cell fate decisions. They focused on early somatosensory neurons as these neurons have been extensively characterized and populations defined, but only key transcriptional components have been identified. Much is unknown about the internal dialogue between cell fate programs that precedes neuronal type decisions. Faure et al. find that indeed competing transcriptional programs are active in maturing neurons that correspond to, in the cases examined here, binary cell fate decisions. This work is important in that it provides extensive context for the key transcriptional components leading to fate splits and identified new (and salient) subtleties these decision points (e.g. Neurod1/2 transient coexpression in Branch B). With revision, this manuscript should be considered for publication.

Major

1. The UMAP of “sensory neuron lineage” is a conglomerate of DRG cells from different ages and different genotypes. While this is a clever approach to target the key neuronal progenitors and populations, how well does this method recapitulate normal lineage? To increase my confidence that this piecemeal approach is representative of normal sensory neuron lineage, I would like to see a side-by-side of this cohort of transcriptomics from multiple lines with a small run of wild-type DRG cells that are not genetically labeled or an explanation in the text why the authors feel this is unwarranted.

By combining single cell RNA sequencing data of traced cells from multiple transgenic mice, our aim was to cover the various developmental steps of the neurogenesis process in the somatosensory system. The advantage of this method is to select enough cells of interest to

process them with smartseq2, enabling deep sequencing necessary for analyzing subtle changes in the dynamic of gene expression. The disadvantage of this strategy is that we are merging data from different mouse strains. Another approach would have been, as suggested by the reviewer, to collect WT cells from different stages. However, this strategy would require thousands of single cells in order to obtain few cells of interest. Indeed, somatosensory neurons have not coalesced into an anatomically defined ganglion before E11.5, hence dissection at earlier time point would cover the whole trunk region of the brachial area, including somites, dermomyotome, skin and mesenchymal cells amongst others. Dissection of the DRG at later stages would still include many non-neuronal cells (including progenitor cells, early glial cells, mesenchymal cells and cells associated with blood vessels) and neurons of the first wave would start representing a very low proportion. Therefore, in case of using WT mice with the need to sequence thousands of cells there is the necessity to use 10X for sequencing such large number of cells, though this methodology give much less resolution in term gene detection compare to Smartseq2 which can detect even low abundance transcripts (in our data set with have in average a detection of 8000 genes per cell). Given the type of computational analysis performed in our study, Smartseq2 was the ideal methodology to use. Hence, we judged that our strategy was a good compromise necessary for obtaining the presented detailed results and its conclusions.

We however understand the concern of the reviewer. We would like to stress out that all the transgenic mouse strains selected have a normal sensory neuron development, including correct timing of sensory neurogenesis and diversification. Also, they do not show any sensory phenotypes. Furthermore, they are on a C57bl6 genetic background or have been back-crossed more than 10 times in a C57bl6 background, limiting the genotype difference. Moreover, the whole bifurcation analysis of the neuronal diversification tree comes from only one mouse line, the *Ntrk3^{Cre}*, which should give confidence in the main analysis and conclusions raised in our study concerning the emergence of neuronal subtypes, which is the main point of our study. To answer the reviewer's point, we now provide new plots in the Supplementary Figure 1 (see below) with, as requested, a side-by-side transcriptomic data from the different mouse strains used in our study.

We also provide a new text in the main manuscript explaining the relevance of the choice of the transgenic lines and the justification of using the multiple tracings strategy to perform computational analysis to study bifurcation, which is described in panel (b).

New text has been added (Results section, line 97 to line 117):

“Differentiation trajectories and major fate splits in the developing somatosensory system. In order to obtain a map of lineage splits from progenitors towards specific sensory subtypes during mouse embryonic development, we sequenced the transcriptome of individual single cells (progenitors and neurons) of the somatosensory neuro-glial progeny of the trunk. For this, we FAC-sorted tdTomato positive (TOM⁺) cells isolated from mouse lines which selectively represent all NCC progenitors $Wnt1^{Cre};R26^{tdTOM}$, $Plp^{CreERT2};R26^{tdTOM}$ (injected with tamoxifen at E11.5 and collected at E12.5) and from neurons specific mouse lines to obtain an enrichment of somatosensory neuronal populations $Isl1^{Cre};R26^{tdTOM}$, $Ntrk3^{Cre};R26^{tdTOM}$ (Fig. supplementary 1a) from E9.5, E10.5, E11.5 and E12.5 and sequenced mRNAs of single cells with high coverage using Smart-seq2 protocol (median of 8070 genes detected per cell) (Fig. 1a and Supplementary Fig. 1c-d; see Methods). The selected stages corresponded to key events of sensory lineage development starting from migrating neural crest (E9.5) and finishing with the end of the second wave of sensory neurogenesis and neuronal subtypes early specification (E12.5). Our dataset contains cells from $Isl1^{Cre};R26^{tdTOM}$ at E9.5 and E10.5 and $Ntrk3^{Cre};R26^{tdTOM}$ at E11.5 and E12.5, both lines label sensory neurons, $Wnt1^{Cre};R26^{tdTOM}$ was used to label neural crest cells (NCCs), glial and neuronal progenitors at E9.5 and E10.5 and $Plp^{CreERT2};R26^{tdTOM}$, to trace progenitors at E11.5. The combination of cells collected after tracing with these Cre lines should provide a complete compendium of neuronal progenitors and early neuronal subtypes in mouse DRG. Co-embedding of the cells from all tracings strategies was made without tracing-based batch correction and lead to overlapping of cells of different tracings following both, a developmental time-wise trajectory and a progression from progenitors to specified neurons (Fig. 1a and Fig. supplementary 1a,b) which together convincingly validate the use of the dataset for further computational analysis.”

We now hope that our explanation has given confidence to our analysis.

2. The authors' use of the *Runx3*^{-/-};*Bax*^{-/-} line was good to support their idea branched paths. However, much of this work has been done previously (Levanon et al., 2002; Inoue et al., 2007; Souichiro et al., 2008). Further, this work shows that a branch point exists but does not require a broad program as the authors conclude. I would like to see an experiment where cultured DRG neurons have genes previously not known to be involved before this study as being involved in pre-bifurcation “prepping” for fate decision be misexpressed, both up and down, with CRISPR or other genetic tools to show that these programs do act as a switch in these fate decisions.

The work referred to by the reviewer indeed shows previous important studies establishing a role for RUNX3 in driving a TRKC⁺ identity and inhibiting the expression of the alternative TRKB receptor within the presumptive proprioceptors. The analysis in our study of the subtle transcriptomic changes in the presumptive proprioceptor in the absence of *Runx3* was not intended to repeat this seminal observation but to provide further understanding and to confirm the role of competing modules prior to cell fate commitment during early differentiation of sensory neurons of the myelinated lineage. Indeed, beside the role of RUNX3 in inhibiting TRKB, which we confirmed here, our data indicate that it also represses gene programs of the alternative fate. Hence, in the absence of RUNX3, not only TRKB is expressed but we observe a shift towards the expression of gene modules associated with the mechanoreceptor fate, which supports the hypothesis of competing programs prior to fate commitment. To better clarify our findings, we have now added a new panel with a scheme recapitulating the experimental steps in Fig. 5 and a new text describing the associated results, which we think will help understanding the strategy and our conclusions.

The second part of the question relates to the observation that the loss of a single, yet important gene (here *Runx3*) is sufficient to change gene programs. The reviewer argues that this could contradict our conclusion on the importance of gene modules for defining cell fate in the somatosensory lineage. However, our data do not indicate that, they instead show that in the absence of an important gene associated with a particular cell fate, another module (associated with an alternative cell fate) can take over. They do not demonstrate that RUNX3 is sufficient to define a proprioceptor fate, but that it is necessary, which is not disproving a combinatorial transcription control of cell fate.

We now have added new text to clarify this in the manuscript, line 356-361: *“Interestingly, these differentiation programs not only activate genes specific of one cell fate but also repress gene programs of the alternative fate. Despite the observation that the loss of RUNX3 in the early myelinated lineage results in a shift of the cell identity which underlines the weight of a single factor in maintaining the integrity of a differentiation program, our computational analysis suggest a combinatorial coding of the cell identity, as previously suggested^{30,31”}*

The *in vitro* approach proposed by the reviewer is interesting but unfortunately non-feasible in our system. Indeed, study using CRISPR or other genetic tools would require massive multiple gene-specific modifications *in vivo* in order to affect the immature neurons prior to fate commitment, which occur as progenitors leave the cell cycle and early post-mitotic neurons start differentiating. This type of biological events, which take less than 24 hours post-progenitor stage are virtually impossible to manipulate in DRG *in vitro*, as somatosensory neurons have not yet coalesced into an anatomically defined ganglion at this stage, hence cannot be dissected and manipulated in a clean culture system. Also, such gene-

modification *in vitro* would most probably affect progenitors as well, leading to phenotypes hardly difficult to decipher. Most determinant, it has been previously shown that culturing DRG neurons changes their pattern of gene expression (Friedel *et al.*, 1997, PNAS), which prevents us from using *in vitro* strategy to study the genetic control of sensory neuron differentiation. Having said that, we tried to investigate the role of some genes (potentially involved in the prepping of cell fate choice), knowing that might not be the perfect experiment either since the fate control is most likely combinatorial. Nevertheless, we tried to get hold on some transcription factor KO mice but unfortunately, with the pandemic situation, collaborators had to close their lab and even get some of their mice colonies cut down.

Minor

1. Not spending a chunk of the discussion contrasting these findings with the new Ginty paper (Sharma *et al.*, 2020) seems like a fairly glaring omission.

We totally agree with the reviewer, but please note that we submitted our manuscript before January, that is prior to the publication of the new Ginty's lab study, explaining its omission in the discussion of the first version of our manuscript. We thank the reviewer to give us the opportunity to discuss these results in our revised version. We have added a new paragraph in the Discussion section, line 486 to line 497:

*“Our analysis also complements a study recently published (Sharma *et al.*, 2020), which presented the transcriptional profile of the somatosensory neurons across various stages covering the embryonic and postnatal development. In contrast to our findings, Sharma *et al.* defined the early postmitotic sensory neurons as “unspecialized””, whereas in our study they already primed towards specific sensory sub-branches. Our data however confirm previous results demonstrating neuronal heterogeneity in DRG as early as E11.5, with clearly defined populations of proprioceptor and mechanoreceptor at this stage (Levanon *et al.*, 2002; Luo *et al.*, 2009; Bourane *et al.*, 2009; Kramer *et al.*, 2006; Wende *et al.*, 2012; Lallemand *et al.*, 2012; Wang *et al.*, 2019). Overall, our studies mostly differ in their focus: Sharma *et al.* opted for analysis of large number of cells for offering a detailed developmental transcriptional atlas of sensory cell types while our study, focusing on fewer early time points, provides a mechanistic insight of the molecular events leading to the early emergence of sensory neuron diversity.”*

Hence, we clearly view the two studies as complementary and think that together both studies will provide a clearer understanding of the molecular events regulating the early emergence and later diversification of sensory neuron types.

2. I can't find how many cells were used. Please state the total number of cells and how many were used for each genotype and each age.

Indeed, we apologize for not having detailed this in the original version of our manuscript. We have now added those numbers in the new Supplementary Fig. 1 (please see the panel provided in response to your main comment #1).

- $Isl1^{Cre}$ at E9.5 (17 cells) and E10.5 (648 cells)
- $Plp1^{CreERT2}$ at E12.5 (129 cells)
- $Ntrk3^{Cre}$ at E11.5 (345 cells) and at E12.5 (350 cells)
- $Wnt1^{Cre}$ at E9.5 (202 cells) and at E10.5 (554 cells)

3. Claims about cell fate bifurcations and lineage seem a bit strong given that much of the information is inferred. Since the data is pooled between FAC-sorted subpopulations from

different genetic lines, it is impossible to know if these splits into different populations are artifacts of the single cells that were enriched for or indicate real lineages. The reporter lines were carefully thought out, but as the authors themselves show, as with the NeuroD1/2 co-expression, there is much we still do not know.

We understand the reviewer's concern. This has now been answered in our response to the main comment #1. Indeed, the whole bifurcation analysis of the neuronal diversification tree comes from only one mouse line, the *Ntrk3^{Cre}*, which should give confidence in the main analysis and conclusions raised in our study concerning the emergence of neuronal subtypes through cell fate bifurcation, which is the main point of our study.

4. While this work does an excellent job proposing that cell fate decisions are antagonizing programs rather than a few key genes, the proposed model in figure 7 is remarkably similar to the previous model for the somatosensory system (Lallemend and Ernfors, 2012). The idea of sequential binary branches for somatosensory development is well established.

We are very pleased to read the reviewer appreciated our work. Concerning the model presented in figure 7, it axes on the first wave only, its aim was to highlight that even the early populations of sensory neurons are emerging from binary decision. We understand that the way it was presented might be confusing so we decided to remove the graph, as we agree with the reviewer that sequential branching have been suggested in the field, for instance in the context of a late diversification of the nociceptive lineage into the peptidergic and non-peptidergic populations of nociceptors (e.g. Gascon *et al.*, 2010) or in the review by Lallemend and Ernfors (2012). Accordingly, we have added new text in the discussion paragraph:

“The notion of branches segregation by mutual repressive activity for somatosensory development had already been suggested for cell types diversification during late embryonic development, for instance during the differentiation of TRKA⁺/RUNX1⁺ nociceptors into a peptidergic and a non-peptidergic population of sensory neurons (Gascon et al., 2010; Lallemend and Ernfors, 2012), yet the timing and gene regulatory strategies that regulate this differentiation process remains poorly understood.”

5. Quantification of the colocalization in fig1i might help to make the point about choice points better

We have now added quantification (see below) and a text that we think might be helpful, Results section, line 163 to line 174:

“From these progenitors, the branch A and branch B differentiated to endpoint neuronal clusters that express by E12.5 the molecular markers characteristic of the main early DRG neuronal populations, the proprioceptive, mechanoreceptive and nociceptive populations. The proprioceptive lineage is characterized at E12.5 by the co-expression of Ntrk3 (Ntrk3 coding for TRKC) with Runx3 (Levanon et al., 2002; Kramer et al., 2006; Wang et al., 2019) while the skin mechanoreceptive neurons express Ret and Ntrk2 (Ntrk2 coding for TRKB) (Fig. 1g-i) (Bourane et al., 2009; Lallemend and Ernfors, 2012). The mechanoreceptors further branches into Ntrk2⁺ only cells and Ret⁺/Ntrk2⁺ cells, which constitute sub-populations of the this lineage with distinct connectivity and function in the adult (Lallemend and Ernfors, 2012; Luo et al., 2009; Li et al., 2011). Altogether, those populations represent the branch A. In contrast, the branch B differentiated into nociceptive Ntrk1⁺ (TRKA) cells and is characterized by the expression of Ntrk1 (and the beginning of induction of Runx1 in the TRKA⁺ cells) (Fig. 1g-i), a specific marker of the nociceptive lineage (Lallemend and

Ernfors, 2012). Combinatorial use of the above cited population markers covers all (ISL1⁺) (Sun et al., 2008) sensory neurons at this stage (Fig. 1i, left panel).

6. It might be nice to somehow incorporate information about migratory waves of NCC into your high dimensional analysis. For example, what happens to the map if you subtract boundary cap cells?

We are grateful to the reviewer for the comment which led us to investigate closely the cluster 3 previously identified as BBCs only. We found that the cluster contains other late NCCs as well, which were hidden in our first analysis due to the selective expression of genes associated to BCCs. We therefore have performed a new clustering of the cluster 2 and cluster 3 individually to explore molecular signature of sub-clusters which we hope will be of interest to many. The new complementary analysis is now in Supplementary figure 2a as follow:

UMAP embedding representation of the single cell RNA sequencing dataset

To answer the reviewer's comment with the removal of the BCCs, the overall structure of the UMAP and diversification tree is similar to the original ones and the main trajectories remain identical.

7. It's not entirely intuitive what fig 1j is showing. Could be more explicit about what the colors mean in the legend or better yet in the panel itself.

This is now done, see below.

Figure legend Fig. 1j. Hierarchical bifurcation model of NCCs-derived sensory neurons differentiation within Branch A based on our scRNAseq data analysis. Mixed color squares reflect the potential fate choice that the lineage retains at the corresponding developmental point.

8. The paper is logically laid out but I think this could use one more pass at line editing. “that remodeling” is duplicated on line 271

It is now corrected.

--

Reviewer #3 (Remarks to the Author):

I generally like single-cell datasets, as they provide a very nice unbiased view on a particular biological process. I do have a few concerns regarding this particular manuscript though:

1) I wish the authors would make a bit more of an effort to share their data with the community. To only deposit raw data on GEO is rather unusual these days. I would have liked for them to at least put up some Excel data for people to download or ideally a file that can be visualised in browsers like loompy or Loupe (though that may be difficult given that this is not 10X).

We are grateful to the reviewer for improving visibility of our work and dataset to the scientific community. As source file, an excel file is provided containing the raw data for each figure panels including the differentially expressed genes for each cluster.

Additionally, all processed data including a pagoda2 web file has been deposited on GEO (GSE150150) with the raw data. To access the files, please use the following code: inwhwayyjfgrcr. The pagoda2 web file (p2w_sensory.bin) can be opened on a browser and allows for exploration of the data in a similar manner as Loupe. To open it, please use the following link:

<http://pklab.med.harvard.edu/nikolas/pagoda2/frontend/current/pagodaLocal/index.html>

Also, we provide the code in form of markdown notebooks for reproducibility and better readability on the following github repository:

https://github.com/LouisFaure/sensoryfates_paper

The repository is for now private and will be made public if and upon acceptance of the manuscript. In the meantime, it is possible to add any reviewer having a github account as a collaborator to access it. The repository files can also be sent by another way upon request.

2) Related to this: if they want their article to be cited and appreciated by regular sensory/developmental biologists, they may want to spend a bit more time explaining how the various algorithms they use work conceptually. How does RNA velocity “reveal a clear directionality” for example? Does it use the temporal nature of the dataset to cluster cells that look similar over time?

More ample clarifications and explanations have been added in the main text, notably about RNA velocity as well as pseudotime related methods (please see below, in response to the action points)

3) Finally, while I appreciate that we are all required to oversell our data slightly to get into journals such as these, I fear the authors may not convince many readers that they have found anything particularly new. This is a point that I don't think requires any action, but just to express my feeling that when I read this, I was trying hard to figure out how exactly this was very different from what has been reported before? Apart maybe from the Neurog1/2 finding (see below). Personally, I do not think this should prevent publication – on the contrary, I regret that we are not permitted to openly celebrate replication and concordance with prior literature in our papers.

Action points:

- Please indicate how you will improve data sharing

Please see answer to the main comment #1.

- Alter your text to explain a bit more about the algorithms you are using and how they work conceptually

The following paragraph has now been added in Results section, line 119 to line 133:

‘We used RNA velocity¹⁰, an unbiased approach that leverages the distinction of spliced and unspliced RNA transcripts from the aligned sequences, allowing us to obtain an additional timepoint for each cells (t : expression of old spliced RNA and $t+1$: expression of new unspliced RNA). Using the proportion between these two values for a given gene, and under some assumptions, it is possible to infer whether a gene is being activated or downregulated. By combining this information for all genes in a given cell, as well as comparing it to its neighbors, it is possible to infer a vector indicating the putative future transcriptional state of the cell. In our dataset, RNA velocity revealed a clear differentiation directionality from neural crest progenitors to neuronal progenies. In order to recapitulate such transitions, the dataset was first projected into diffusion space to denoise the underlying geometry, and a principal tree has been fitted using ELPiGraph¹¹ in a semi-supervised way with the help of clustering results (Fig. 1b,c). ELPiGraph is a manifold learning method which aims at inferring a principal graph (such as a tree) ‘passing through the middle of data’ in high dimension. Cells were ordered along the principal tree, and for each cell the distance on the graph to a chosen root is considered as a pseudotime value’.

- Add your FACS gating strategy to the supplementary and add details in the methods as to how you FACSeD (e.g. did you use a live/dead stain? What did you sort into, etc)

Details have been added to the material and method section, line 801 to line 811.

“Single TOMATO+ cells were sorted by fluorescence-activated cell sorting (FACS) into individual wells containing lysis buffer in a 384-well plate provided by the single cell Facility, Karolinska Institutet. The plates were immediately placed on dry ice and stored at -80°C before processed for Smart-seq2 protocol.”

- Explain exactly how you batch controlled these experiments – this is of course of particular relevance given that you were examining differential expression over time.

In the methods is mentioned the existence of a batch effect correction (line 835 to line 837):

“A biological aspect linked to mitochondrial respiratory chain has been identified to be different among batches from the same condition. This aspect was regressed on the raw count matrix using ScaleData from Seurat package”

No other corrections were needed as different genotypes and timepoints do overlaps in transitions that makes sense biologically (see new genotype plots, Suppl. Figure 1a).

- State somewhere clearly that you didn't use UMIs and how this affects data interpretation (i.e. you may have PCR duplicates in your data?)

We have now specified that SmartSeq2 does not use UMI.

We also added a new paragraph in the material and methods section (line 848 to line 851) describing a count matrix correction step prior to the pseudotime analysis. Indeed, the count matrix has been modelled using scde R package. scde fits individual error models for single cells using counts derived from single-cell RNA-seq data to estimate drop-out and amplification biases on gene expression magnitude.

- Provide some kind of quantification for Figures 2e&f. Right now, I'm not overly convinced.

We now provide quantifications for previously named Fig2e and 2f (please see below).

We have performed a new experiment where the Neurog2^{CreERT2};Rosa26^{tdTOM} pregnant mice were injected with tamoxifen at E9.5 and E10.5 to cover both waves of neurogenesis. The embryo were collected at E18.5. Using this strategy, we would expect to trace the neurons born between E10 and E12, those born before being Tomato positive and after that period being tomato negative. Our results confirm this, with traced cells being found in both large diameter neurons (expected) and in large proportion in small diameter TRKA neurons (of the second wave), confirming our previous data. We have now provided quantification (see panel below). For the Neurog1cre;R26^{tdTom}, our data also confirm data recently published (during the review process of the current manuscript) by Sharma et al. (2020, Nature).

Figure legend: **d**, Analysis of the neurogenic niche with RNA Velocity computation showing sequential expression of the main neurogenic transcription factors *Neurog2* and *Neurog1*. **e**, RNAscope staining for *Neurog1* and *Neurog2* on E10.5 DRG sections confirms their co-expression in the same progenitors. **f**, Plot of single cells values for *Neurog1* and *Neurog2* shows the existence of 3 stages among progenitors following the pseudotime at E10.5, including concomitant expression of *Neurog2* and *Neurog1* at the single cells level (within dash lines, 136 cells). **g**, Quantification of the concomitant average expression among neuronal progenitor cells reflect a dynamic range of expression from high to low *Neurog2* expression and low to high expression of *Neurog1*. **h**, Cross-section of E18.5 DRG from *Neurog2^{CreERT2};R26^{tdTom}* injected at E9.5 and E10.5 with tamoxifen shows recombination in neurons originating from the two waves of neurogenesis (branches A and B), as shown by expression of TOM in large diameter neurons and in small diameter TRKA positive neurons (asterisks) (arrows point to TRKA⁺/RFP⁻ cells). **i-k**, 533 TOM+ cells were analyzed per animal (**i**, n=4). Among the RFP+ neurons, more than half were TRKA positive (**j**), with a diameter inferior or equal to 15µm (**k**). Scale bar, 20µm. **l**, Similar to (**h**), using *Neurog1^{Cre};R26^{tdTom}*, we show recombination (TOM positivity) within neurons of the 2nd wave (NF200⁻) and a majority of neurons originating from the 1st wave of neurogenesis, as identified by NF200 positivity.

Reviewers' Comments:

Reviewer #1:

Remarks to the Author:

The authors have adequately addressed the concerns raised in my original review.

Reviewer #2:

Remarks to the Author:

Very impressive revision. All my concerns have been addressed. Recommend publication.

Reviewer #3:

Remarks to the Author:

I'd like to thank the authors for their extensive reply to my comments. I also really appreciate the extra efforts they have now made to facilitate data sharing. I have no further concerns and would be very pleased to read such a nice paper in Nature Communications.

Responses to reviewer comments

REVIEWERS' COMMENTS:

Overall, we are thankful to the reviewers for improving via their comments our manuscript and for describing our manuscript as acceptable for publication in Nature Communications. Please see answer in blue.

Reviewer #1 (Remarks to the Author):

The authors have adequately addressed the concerns raised in my original review.

We are glad the reviewer is satisfied with our revision.

Reviewer #2 (Remarks to the Author):

Very impressive revision. All my concerns have been addressed. Recommend publication.

We feel encouraged by such comments, thank you. We are happy the reviewer is satisfied with the revision.

Reviewer #3 (Remarks to the Author):

I'd like to thank the authors for their extensive reply to my comments. I also really appreciate the extra efforts they have now made to facilitate data sharing. I have no further concerns and would be very pleased to read such a nice paper in Nature Communications.

We are glad the reviewer notice our effort to facilitate data sharing and would like to thank the reviewer for it. We are happy the reviewer is satisfied with the revision.